# The oxidative fumarase FumC is a key contributor for E. coli fitness under iron-limitation and during UTI

**Stephanie D. Himpsl**[1☯], **Allyson E. Shea**[1☯], **Jonathan Zora**[1¤], **Jolie A. Stocki**[1], **Dannielle Foreman**[2], **Christopher J. Alteri**[1,2]*, **Harry L. T. Mobley**[1]

**1** Department of Microbiology and Immunology, University of Michigan Medical School, Ann Arbor, Michigan, United States of America, **2** Department of Natural Sciences, University of Michigan Dearborn, Dearborn, Michigan, United States of America

☯ These authors contributed equally to this work.
¤ Current address: University of Detroit Mercy School of Dentistry, Graduate Periodontics Program, Detroit, Michigan, United States of America
* alteri@umich.edu

## Abstract

The energy required for a bacterium to grow and colonize the host is generated by metabolic and respiratory functions of the cell. Proton motive force, produced by these processes, drives cellular mechanisms including redox balance, membrane potential, motility, acid resistance, and the import and export of substrates. Previously, disruption of succinate dehydrogenase (*sdhB*) and fumarate reductase (*frdA*) within the oxidative and reductive tri-carboxylic acid (TCA) pathways in uropathogenic *E. coli* (UPEC) CFT073 indicated that the oxidative, but not the reductive TCA pathway, is required for fitness in the urinary tract. Those findings led to the hypothesis that *fumA* and *fumC* encoding fumarase enzymes of the oxidative TCA cycle would be required for UPEC colonization, while *fumB* of the reductive TCA pathway would be dispensable. However, only UPEC strains lacking *fumC* had a fitness defect during experimental urinary tract infection (UTI). To further characterize the role of respiration in UPEC during UTI, additional mutants disrupting both the oxidative and reductive TCA pathways were constructed. We found that knock-out of *frdA* in the *sdhB* mutant strain background ameliorated the fitness defect observed in the bladder and kidneys for the *sdhB* mutant strain and results in a fitness advantage in the bladder during experimental UTI. The fitness defect was restored in the *sdhBfrdA* double mutant by complementation with *frdABCD*. Taken together, we demonstrate that it is not the oxidative or reductive pathway that is important for UPEC fitness *per se*, but rather only the oxidative TCA enzyme FumC. This fumarase lacks an iron-sulfur cluster and is required for UPEC fitness during UTI, most likely acting as a counter measure against exogenous stressors, especially in the iron-limited bladder niche.

**Data Availability Statement:** All relevant data are within the manuscript and its Supporting Information files.

**Funding:** This work was supported by Public Health Service grant AI059722 from the National Institutes of Health to HM. The funders had no role in study design, data collection and analysis, decision to publish, or preparation of the manuscript.

**Competing interests:** The authors have declared that no competing interests exist.

## Author summary

All living organisms have the ability to harvest energy from their environment. Many organisms, including microbes such as *E. coli*, use cellular respiration to release energy from the chemical bonds in sugars and proteins. The tricarboxylic acid (TCA) cycle and the electron transport system are intimately linked to each other and to the process of converting the released energy into the ability to do work. In this case, the work being explored in this study demonstrates how this process of cellular respiration contributes to the ability for uropathogenic *E. coli* to colonize the urinary tract and cause infection. Using a genetic approach, we found that certain TCA cycle enzymes together, such as succinate dehydrogenase and fumarate reductase, are dispensable for *E. coli* during UTI, while the oxidative fumarase FumC is absolutely required for UTI. These findings suggest that modularity in respiratory or TCA cycle components are important for *E. coli* fitness during infection and could reflect an important strategy to subvert host defenses rather than replication in an aerobic or anaerobic environment.

## Introduction

The host gastrointestinal and urinary tract are two distinct host microenvironments that uropathogenic *Escherichia coli* (UPEC) uniquely navigate and colonize by using specific virulence properties and metabolic pathways [1–6]. The nutrient-rich gastrointestinal tract is a highly competitive environment for a large number and diverse set of bacterial species. Conversely, the nutrient-limited urinary tract is a less competitive niche for bacteria to colonize due to highly vigilant host defense mechanisms. Previously, we demonstrated that of the central metabolism pathways only gluconeogenesis and the tricarboxylic acid (TCA) cycle are essential for UPEC fitness within the urinary tract, in contrast with the metabolic requirements for intestinal colonization of commensal *E. coli* [7]. Glycolysis and the pentose phosphate pathway are dispensable during extraintestinal infection of *E. coli*, while peptide import, gluconeogenesis, and the TCA cycle are required for uropathogenesis [1, 5, 8]. These findings indicate that the enzymes needed to catabolize sugar are not required for *E. coli* fitness, while gluconeogenesis, which functions to convert amino acids and other gluconeogenic substrates into sugars, is crucial for survival in the urinary tract. This is perhaps, in part, due to urine serving as the medium in the urinary tract environment, which provides a dilute mixture of peptides and amino acids as the main carbon source [9]. In addition to releasing stored energy in the form of ATP through aerobic oxidation of acetyl-CoA and pyruvate, the TCA cycle provides precursors of amino acids as well as NADH that is transferred to the electron transport pathway for the re-oxidation to $NAD^+$ which is essential for multiple cellular functions. Previously, we reported that the oxidative TCA cycle is required for UPEC fitness [1, 5], providing additional evidence that the mouse urinary tract is a moderately oxygenated environment [10].

For bacteria to persist in the host they must be able to grow and adapt to their environment. To accomplish these tasks, facultative bacteria systematically employ modular respiratory systems to meet the energy and redox demand for survival in various environments. When inhabiting a new environment, bacteria optimize their growth by substituting and altering available respiratory components within a vast modulatory network. This bacterial respiratory network is intricate yet redundant, consisting of multiple substrate-specific dehydrogenases needed to carry out oxidation of reduced electron carriers such as NADH, mobile quinone pools that deliver electrons to terminal oxidoreductases, and a diverse assortment of terminal electron acceptors [11]. The ability of bacteria to sense and respond to the environment via respiration

is not only essential for growth but also for multiple cellular mechanisms including; redox balance, membrane potential, motility, acid resistance, and the import and export of substrates. This respiratory heterogeneity has been suggested to play a role in UPEC pathogenesis [12]. These mechanisms are driven by the energy derived from the proton motive force generated by respiratory systems that depend upon reactions preformed within the TCA cycle (Fig 1). A complete aerobic TCA cycle oxidizes carbon skeletons and concomitantly produces relatively large pools of the reduced electron carrier NADH. This NADH must be re-oxidized to $NAD^+$ by donating electrons to an electron transport chain and subsequently protons are translocated across the cytoplasmic membrane. During anaerobic conditions, bacteria must limit the amount of NADH produced due to the reduced efficiency to re-oxidize it, yet still require TCA cycle intermediates as carbon skeletons for cellular functions like amino acid biosynthesis. To circumvent the inability to efficiently re-oxidize $NAD^+$ during anaerobic conditions, bacteria implement a branched, reductive TCA pathway (Fig 1) coupled with other anaerobic respiration or fermentation pathways.

Equipped with an arsenal of components to carry out oxidative and reductive respiration, facultative bacteria like *E. coli* are readily capable of adapting their physiology and metabolic behavior to benefit their survival in a variety of host environments, such as the intestine and urinary tract. These host environments can be fluid in the sense that microaerophilic niches may shift to more anaerobic ones and vice versa. To colonize the mouse gastrointestinal tract, enterohemorrhagic *E. coli* (EHEC) and commensal *E. coli* both demonstrate this metabolic flexibility by taking advantage of the exogenous electron acceptors, oxygen and nitrate, in the host [13]. Interestingly, colonization within the same host environment by *Salmonella enterica serovar* Typhimurium only requires the oxidative TCA cycle [14, 15] and the inflammatory environment may potentially liberate other alternative electron acceptors such as tetrathreoniate in the gut [16]. Previously, we discovered that in the mouse urinary tract the oxidative TCA cycle is important for UPEC fitness while the branched, reductive TCA pathway is dispensable [1, 5]. Disruption of succinate dehydrogenase, *sdhB*, in the oxidative TCA cycle resulted in a fitness defect in mice during cystitis [1, 17], while disruption of the reductive TCA pathway with a mutation in fumarate reductase, *frdA*, resulted in a fitness advantage in the mouse bladder [5]. In addition, loss of the alternative fumarase, *fumC*, which also functions in the complete, oxidative TCA cycle causes a fitness defect in the bladder of mice during experimental UTI [5]. These findings support previous evidence that the urinary tract is a moderately oxygenated environment [10]; however, *Proteus mirabilis*, which colonizes the same niche as UPEC, requires both the oxidative and reductive TCA pathways suggesting that an anaerobic microenvironment may exist within this host environment [5]. Because sensing of oxygen depletion is important during *E. coli* UTI fitness [18], it is possible that loss of the reductive TCA pathway enzyme subunit FrdA can be compensated for by the presence of alternative energy pathways like Ni-Fe hydrogenases and nitrate reductases [19]. In addition, succinate dehydrogenase (SDH) and fumarate reductase (FRD) substitute for one another to catalyze the interconversion of fumarate and succinate in both the oxidative and reductive TCA pathways [20–23]. Further investigation of SDH and FRD during *in vivo* UPEC colonization of the urinary tract may be beneficial in deciphering the roles of oxidative and reductive TCA pathways.

A drawback from our previous study, in which metabolic pathways were systematically disrupted, was the limited investigation of two additional fumarase genes encoded by *E. coli*. All three fumarases are genetically and biochemically distinct and catalyze the interconversion of fumarate and malate in the TCA cycle. Fumarase FumA catalyzes the same oxidative TCA cycle step as FumC, converting fumarate into malate under presumably aerobic conditions. The third fumarase, FumB, operates in the reductive branched pathway during anaerobic

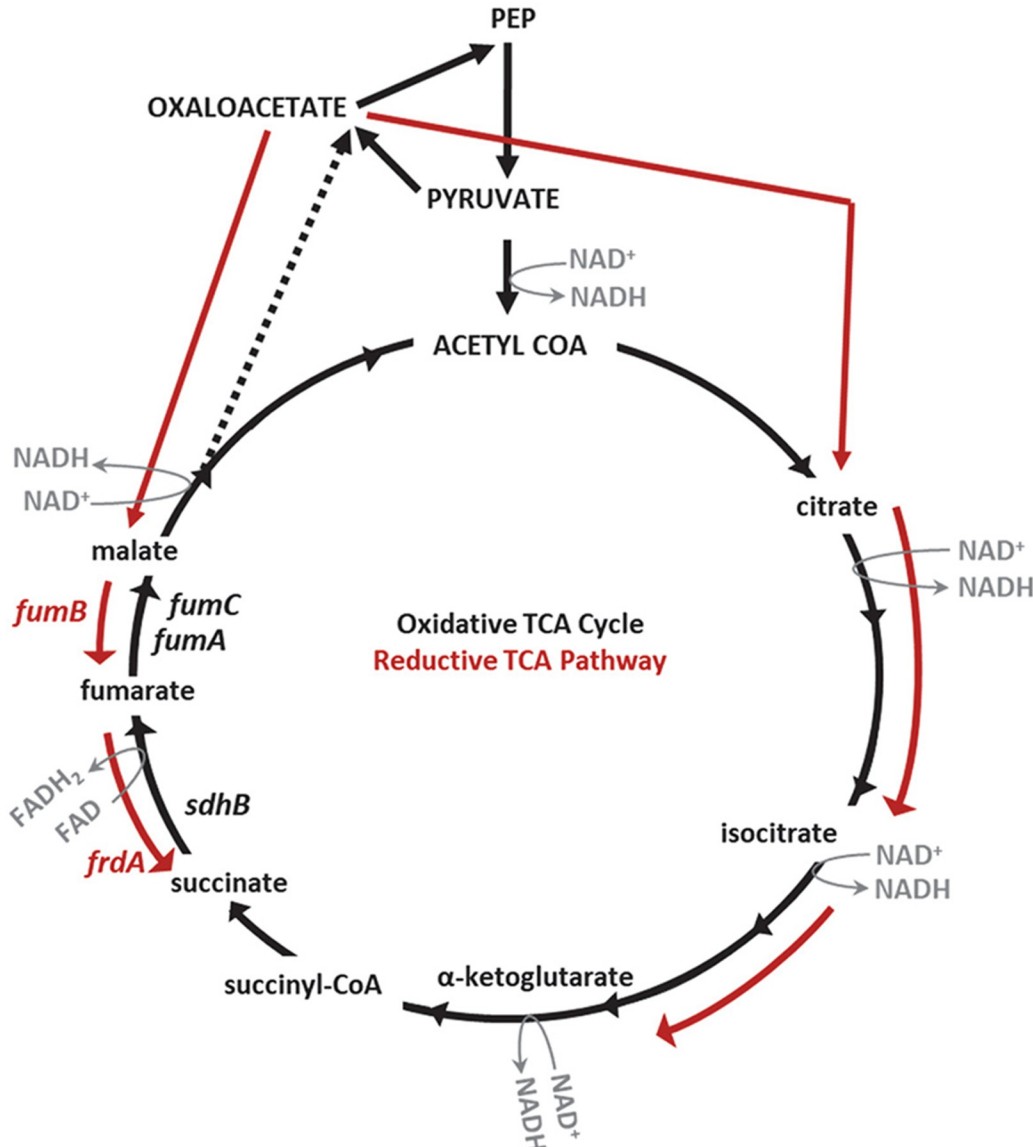

**Fig 1. Diagram of the oxidative and reductive TCA pathways.** Reactions and targeted genes (*sdhB*, *fumA*, *fumC*) of the oxidative TCA cycle are denoted by black arrows and black font. Reactions and targeted genes (*fumB*, *frdA*) of the branched, reductive TCA pathway are denoted by red arrows and red font. NADH and FADH$_2$ are generated during reactions of the oxidative TCA cycle. Arrows indicate the direction of each reaction.

conditions and catalyzes the conversion of malate into fumarate. Previously, a *fumB* mutation in *Salmonella serovar* Typhimurium SR-11 was found to play no role in virulence within the intestine; however, a mutation in *fumAC* was attenuated but not avirulent, suggesting that either FumA or FumC or both play a role during infection [24]. Based upon our previous findings in which FumC and SDH was required for UPEC infection [1, 5], we hypothesized that FumA would be essential for colonization while FumB would be dispensable. To address the roles of these additional fumarase enzymes during colonization of the urinary tract we constructed additional single mutant strains *fumA* and *fumB*, double mutant strains *fumAB*, *fumAC*, and *fumBC*, and triple mutant strain *fumABC* in the parental strain, CFT073, and examined these during experimental UTI as well as in *in vitro* experiments. Any double

mutant strain containing a disruption of *fumC* and the triple mutant strain *fumABC* resulted in a fitness defect in both the bladder and kidneys of mice during UTI. As expected, disruptions within reductive branched pathway enzyme FumB had no effect on UPEC fitness. Surprisingly, FumA of the oxidative TCA cycle was not required for UPEC infection. To further investigate the role of both the oxidative and reductive TCA pathways during UTI, an additional mutant strain, *sdhBfrdA*, was constructed in which the same step in both the oxidative and reductive pathway of the TCA cycle were disrupted. The mutation of *frdA* in the *sdhB* background shifted the fitness defect observed with the loss of *sdhB* alone into a fitness advantage in the bladder during experimental UTI. These findings suggest that UPEC survival within the urinary tract is not simply a question of whether an oxidative or reductive TCA pathway is required, for example as with oxygen-dependence, but perhaps other host-imposed conditions including iron-limitation and/or oxidative stress are driving the *E. coli* decisions regarding how to modulate energy metabolism during infection. To help decipher the underlying mechanism for our *in vivo* UTI findings, we carried out additional experiments to examine the TCA cycle mutants during iron-limitation, oxidative stress using hydrogen peroxide, as well as acid resistance and antibiotic susceptibility.

## Results

### FumC is required for *E. coli* fitness *in vivo*

The fumarase, FumC, catalyzes the oxidative TCA cycle step by which fumarate is converted to malate (Fig 1). Previously, we have shown that deletion of *fumC* in *E. coli* CFT073 results in a fitness defect in the mouse bladder during urinary tract infection (UTI), supporting the hypothesis that the oxidative TCA cycle plays a role during infection [1, 5]. *E. coli* encodes an additional fumarase, *fumA*, adjacent to *fumC* in the CFT073 genome. Like FumC, FumA also catalyzes fumarate oxidation to malate during the oxidative TCA cycle, meaning that these enzymes can both perform the same reaction with equal substrate affinities [25] (Fig 1). A third fumarase, *fumB*, is encoded by *E. coli*; however, unlike FumA and FumC, FumB operates in the branched, reductive TCA pathway converting malate to fumarate (Fig 1). Despite operating in different TCA cycles, FumA and FumB have 90% identity to each other. To further investigate the *in vivo* roles of these additional fumarase enzymes, single mutant strains *fumA* and *fumB*, double mutant strains *fumAB*, *fumAC*, and *fumBC*, and a triple mutant strain *fumABC* were constructed in CFT073 and examined during experimental UTI. Based upon our previous findings, we hypothesized that the mutant strains with disruptions within the oxidative TCA cycle (*fumA* and/or *fumC*) would result in a fitness defect following infection while mutant strains with a disruption in the reductive TCA pathway (*fumB*) would be dispensable during UTI.

To directly test each individual fumarase enzyme, double mutant strains were constructed in which only one fumarase is present; double mutant strain *fumAB* only encodes *fumC*, double mutant strain *fumAC* only encodes *fumB*, and double mutant strain *fumBC* only encodes *fumA*. Interestingly, we observed that following co-challenge with wild-type CFT073, mutant strain *fumAB* displayed a fitness advantage in the bladder ($P = 0.0011$) (Fig 2A). In contrast, co-infection with wild-type CFT073 and double mutant strain *fumAC* or *fumBC*, both demonstrated a significant fitness defect in the bladder ($P = 0.0002$ and $P = 0.0001$, respectively) and kidneys ($P = 0.0011$ and $P = 0.0067$, respectively) (Fig 2B and 2C). Consistent with this, disruption of all three fumarases in mutant strain *fumABC* also resulted in a significant fitness defect in the bladder and kidneys ($P = 0.0005$ and $P = 0.0012$, respectively) (Fig 2D). These findings clearly indicate that any mutant strain with a disruption in the *fumC* gene (*fumC*, *fumAC*,

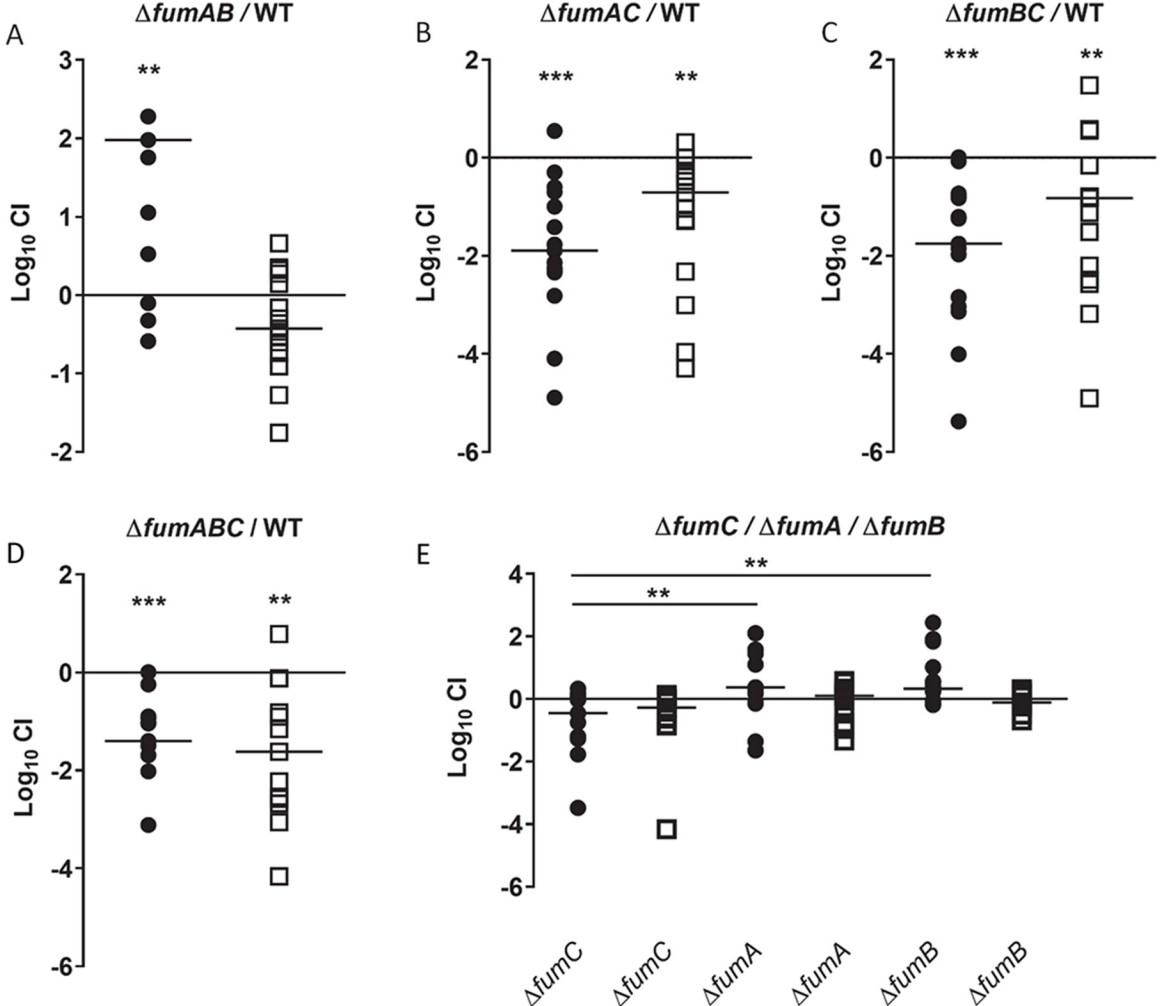

**Fig 2. FumC is required for *in vivo* fitness of CFT073.** *In vivo* Log$_{10}$ competitive indices (CI) were determined following 48 h co-challenge infections in female CBA/J mice with a 1:1 ratio of (A) double mutant *fumAfumB* and wild-type CFT073, (B) double mutant *fumAfumC* and wild-type CFT073, (C) double mutant *fumBfumC* and wild-type CFT073, and (D) triple mutant *fumAfumBfumC* and wild-type CFT073. (E) *In vivo* Log$_{10}$ CI was determined following infection with a 1:1:1 ratio of single mutants *fumC*, *fumA* and *fumB*. Each symbol represents bladder (closed circle) and kidneys (open square) from an individual mouse. Log$_{10}$ CI>1 indicates a fitness advantage and Log$_{10}$ CI<1 indicates a fitness defect of the mutant. Significant differences in colonization (**$P<0.01$, ***$P<0.001$) were determined with the Wilcoxon signed-rank test. Triple co-challenge data was analyzed using a one-way ANOVA with multiple comparisons (**$P<0.01$).

*fumBC*, and *fumABC*) results in a fitness defect during infection suggesting that only *fumC* is required for UTI.

To further investigate the individual fumarase single mutants *in vivo* and rank their fitness in relationship to each other, an infection with all three single mutants *fumA*, *fumB*, and *fumC* in a 1:1:1 ratio was examined during experimental UTI. Only mutant strain *fumC* was statistically outcompeted by mutant strain *fumA* ($P = 0.0041$) and mutant strain *fumB* ($P = 0.0022$) in the mouse bladder suggesting loss of FumC is detrimental to UPEC UTI (Fig 2E). Indeed, a fitness advantage during experimental UTI was only observed for double mutant strain *fumAB* which is able to encode *fumC*. These findings suggested that only FumC of the oxidative TCA cycle is important for CFT073 colonization of the mouse urinary tract. Additionally, these

results also suggest that eliminating non-essential fumarase enzymes, FumA and FumB, may enhance fitness for UPEC in the bladder microenvironment.

## Loss of FRD ameliorates the fitness defect in the SDH mutant

Membrane-bound succinate dehydrogenase (SDH) consists of four subunits, encoded by *sdhCDAB*, which catalyzes succinate oxidation to fumarate during the oxidative TCA cycle and donates electrons to ubiquinone by being a component of the aerobic respiratory chain (Fig 1). In the branched, reductive TCA pathway fumarate reductase (FRD), encoded by *frdABCD*, catalyzes the reduction of fumarate to succinate and also can function in the terminal step of electron transfer to fumarate during anaerobic respiration when fumarate is available (Fig 1). Previously, disruption of SDH resulted in a significant fitness defect during experimental UTI [1, 17] and loss of FRD did not result in a fitness defect [5] suggesting that the oxidative TCA cycle is required for UTI while the reductive TCA pathway is dispensable [1]. Because only one of the two oxidative fumarase isozymes (FumC) was required for UPEC fitness, we decided to more thoroughly investigate the roles of oxidative and reductive respiration during experimental UTI.

*In vivo* co-challenge between single mutant strain *sdhB* and single mutant strain *frdA* resulted in a fitness defect of the *sdhB* mutant in both the bladder and kidneys ($P = 0.0004$ and $P = 0.0413$), providing additional evidence that the oxidative TCA cycle is crucial during UTI (Fig 3A). Interestingly, loss of FRD in the *sdhB* mutant background (*sdhBfrdA* double mutant strain) ameliorates the fitness defect observed for the mutant *sdhB* strain [1] and generates a fitness advantage in the bladder ($P = 0.0105$) (Fig 3B). These findings indicate that FRD, in the absence of SDH, creates a fitness defect *in vivo*. Indeed, when an *in vivo* co-challenge was performed between the *sdhBfrdA* double mutant and single mutant strain *frdA* no significant difference was observed between the strains (Fig 3C). To confirm that the fitness advantage of the *sdhBfrdA* double mutant strain was due to the loss of FRD, the *frdABCD* operon was expressed in the *sdhBfrdA* double mutant strain and the fitness advantage was lost and, moreover, complementation produced a fitness defect during UTI in the bladder and kidneys ($P = 0.002$ and $P = 0.0117$, respectively) (Fig 3D) reminiscent of the fitness defect observed in the *sdhB* single mutant. These experiments suggest that SDH may be important for UPEC UTI, but the presence of FRD has a more significant detrimental effect during infection. However, the fitness defect observed for the *sdhB* mutant strain during co-infection with *frdA* mutant strain, suggests that succinate oxidation is conditionally beneficial to infection when functional FRD is present. Only when the *sdhBfrdA* mutant strain is complemented with FRD do we observe that fumarate reduction is detrimental during infection in the absence of succinate oxidation. Here we propose that FRD is detrimental to UPEC during UTI but can only be observed in the absence of SDH.

## The presence of at least one fumarase enzyme is required for *in vitro E. coli* growth

To determine if the fumarase enzymes are required for growth *in vitro*, growth of single mutant strains *fumA* and *fumB*, double mutant strains *fumAB*, *fumAC*, and *fumBC*, and triple mutant strain *fumABC* were measured in LB medium, defined medium containing either 0.2% glycerol or 0.2% glucose as the sole carbon source, and human urine which mimics the milieu of the host urinary tract. Only triple mutant strain *fumABC* displayed a minor growth defect in LB medium (Fig 4A), a slightly more drastic growth defect in defined medium containing 0.2% glycerol (Fig 4B), a modest growth defect in defined medium containing 0.2% glucose (Fig 4C), as well as an observable growth defect in human urine (Fig 4D) compared to wild-

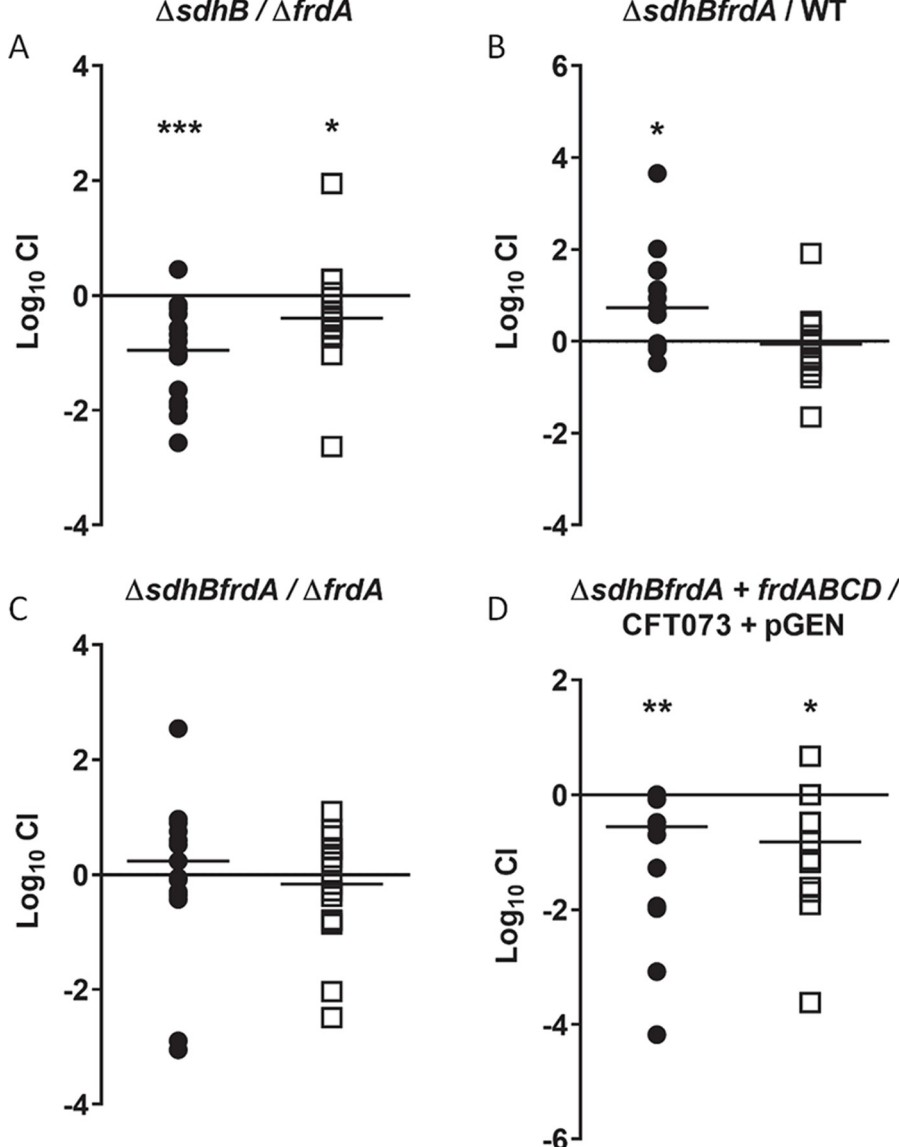

**Fig 3. Presence of FRD in the absence of SDH, creates a fitness defect *in vivo*.** *In vivo* Log₁₀ competitive indices (CI) were determined following 48 h co-challenge infections in female CBA/J mice with a 1:1 ratio of (A) single mutant strain *sdhB* and single mutant strain *frdA*, (B) double mutant *sdhBfrdA* and wild-type CFT073, (C) double mutant strain *sdhBfrdA* and single mutant strain *frdA* and (D) double mutant strain *sdhBfrdA* complemented with pGEN-*frdABCD* and wild-type CFT073 containing pGEN-MCS. Each symbol represents bladder (closed circle) and kidneys (open square) from an individual mouse. Log₁₀ CI>1 indicates a fitness advantage and Log₁₀ CI<1 indicates a fitness defect of the mutant. Significant differences in colonization (*$P<0.05$, **$P<0.01$, ***$P<0.001$) were determined with the Wilcoxon signed-rank test.

type CFT073. These observations suggested that the presence of at least one fumarase enzyme is required for growth in multiple aerobic media conditions. The *in vitro* growth defect of triple mutant strain *fumABC* was restored to the wild-type phenotype when complemented with pGEN-*fumC* and cultured in LB medium as well as defined medium containing 0.2% glycerol or 0.2% glucose (S1A, S1B and S1C Fig respectively). The poor *in vitro* growth observed for the triple mutant strain *fumABC* may contribute to the *in vivo* fitness defect observed during infection (Fig 2D); however, the single and double mutant strains with disruption in *fumC* which

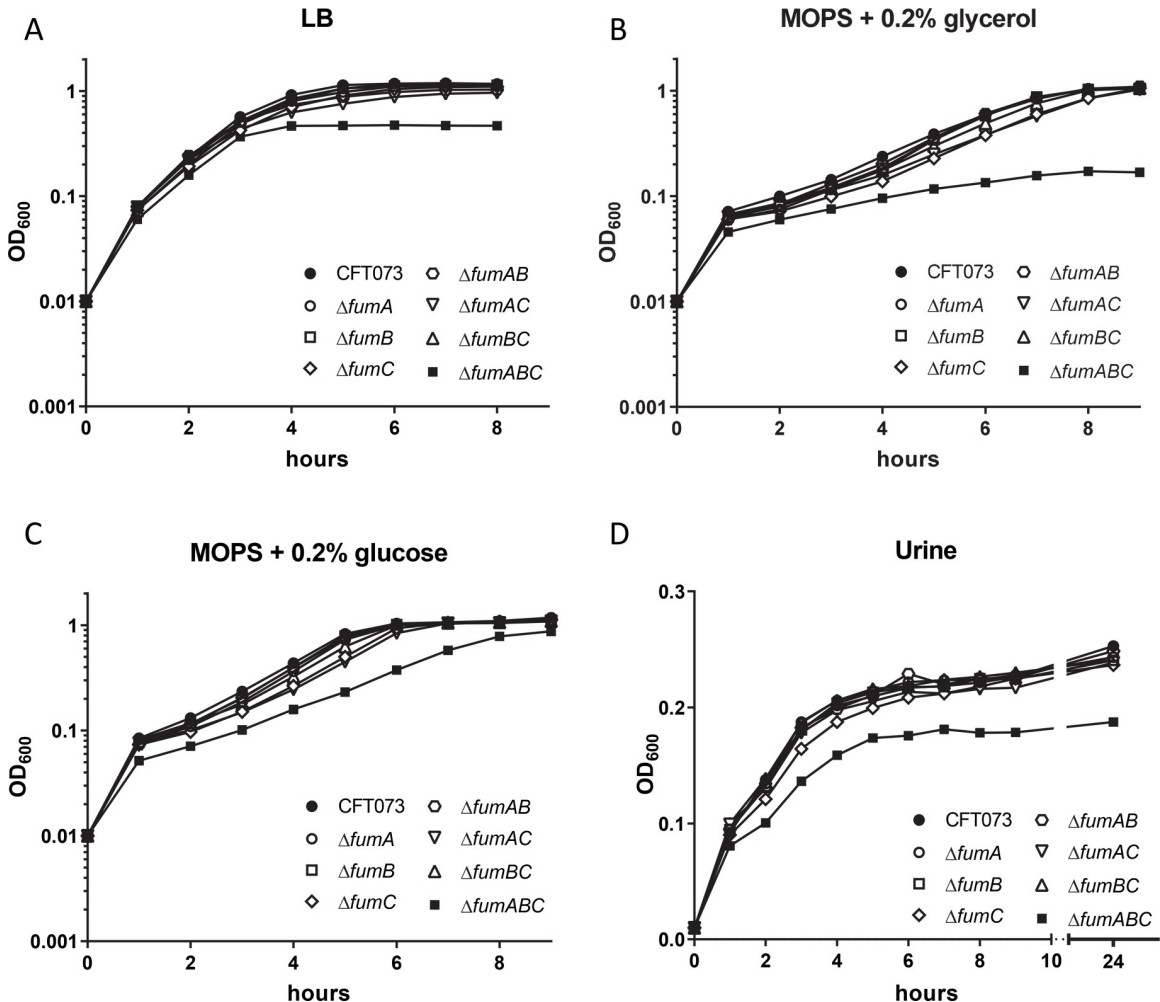

**Fig 4. Optimal *E. coli* growth *in vitro* requires the presence of at least one fumarase.** Growth of wild-type CFT073, fumarase single mutant strains *fumA*, *fumB*, and *fumC*, double mutant strains *fumAB*, *fumAC*, *fumBC* and triple mutant strain *fumABC* in (A) LB medium, (B) defined medium containing 0.2% glycerol as the sole carbon source (C) defined medium containing 0.2% glucose as the sole carbon source, and (D) human urine. OD$_{600}$ values were recorded each hour and the mean of three independent trails is plotted.

also resulted in *in vivo* fitness defects (Fig 2B, 2C and 2E) displayed no deficiency in any of the growth mediums tested. Therefore, generalized growth defects do not demonstrate a potential mechanism for loss of *in vivo* fitness for mutants lacking *fumC*.

## SDH is required for *in vitro* growth of *E. coli*

To determine the roles of SDH and FRD *in vitro*, the single mutant strains, *sdhB* and *frdA*, and double mutant strain, s*dhBfrdA*, were cultured in LB medium, defined medium containing either 0.2% glycerol or 0.2% glucose as the sole carbon source, and human urine. Strains that were unable to express SDH; single mutant strain *sdhB* and double mutant strain *sdhBfrdA*, had a very slight growth defect in LB medium compared to wild-type CFT073 (Fig 5A). Single mutant strain *sdhB* was also observed to have a slight growth defect in defined medium containing 0.2% glycerol while the double mutant strain *sdhBfrdA* had the most pronounced defect (Fig 5B). Both growth defects in LB medium and defined medium containing 0.2% glycerol were complemented with expression of *sdhB* in the single mutant *sdhB* and double mutant

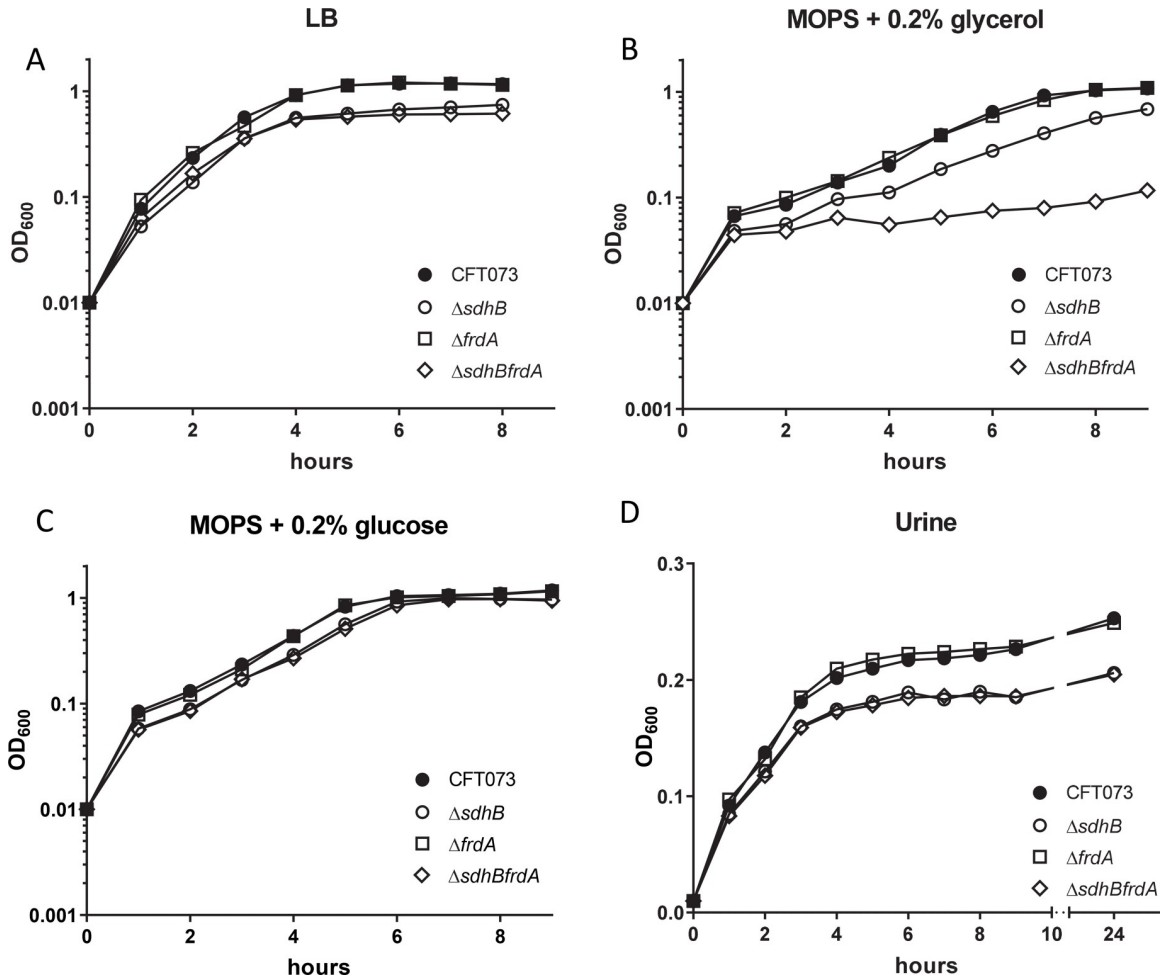

**Fig 5. *In vitro* growth for *E. coli* SDH and FRD mutant strains.** Growth of wild-type CFT073 and succinate dehydrogenase mutant strain *sdhB*, fumarate reductase mutant strain *frdA*, and double mutant strain *sdhBfrdA* in (A) LB medium, (B) defined medium containing 0.2% glycerol as the sole carbon source, (C) defined medium containing 0.2% glucose as the sole carbon source, and (D) human urine. Mean OD$_{600}$ values determined at each hour time point are plotted against time.

*sdhBfrdA* (S2A and S2B Fig, respectively). All mutant strains had a similar growth phenotype as observed for the wild-type strain in defined medium containing 0.2% glucose (Fig 5C). The single mutant strain *sdhB* and the double mutant strain *sdhBfrdA* both had a growth defect in human urine (Fig 5D) similar to the defect observed in LB medium (Fig 5A). These results are not surprising since LB medium and human urine both consist of peptides and amino acids. While the *in vivo* fitness defect of the single mutant strain *sdhB* (Fig 3A) can be partially explained by the growth defects observed *in vitro*, the loss of both *sdhB* and *frdA* in the double mutant strain was found to have no fitness defect *in vivo* (Fig 3B) and cannot explain its poor ability to grow *in vitro*. These findings suggest that under the conditions tested, which were aerobic, SDH is required for optimal growth of *E. coli in vitro*.

## Loss of FumC results in delayed growth during iron-limitation

Because the urinary tract is an iron-limited environment and iron is essential for UPEC pathogenesis, we sought to determine the effect of disruptions within the oxidative and reductive TCA pathways during iron-limitation *in vitro*. Previously, iron availability in *E. coli* has been

shown to be important for expression of respiratory genes [26]. During iron-limitation, expression of aerobic respiratory genes were found to be upregulated while expression of anaerobic respiratory genes including *frdABCD* was downregulated [26]. Iron control has not been observed for SDH [27]. Classified as a class II enzyme, FumC, lacks an iron-sulfur cluster and does not require iron for its activity [28] while class I enzymes, FumA and FumB, are iron-dependent hydrolases [29]. During iron-restricted conditions *fumC* was observed to have increased expression as *fumA* maintained constant levels of expression suggesting a functional FumA requires iron for assembly of the 4Fe-4S iron-sulfur center [30]. Because FumA and FumC both carry out the same reaction within the oxidative TCA cycle, it has been proposed that FumC acts as an alternative enzyme substituting for FumA during iron-limitation and oxidative stress [30].

To mimic the iron-limited environment of the urinary tract, all respiration mutant strains were cultured in iron-free minimal medium with 0.2% glucose as the carbon source, with and without 36 μM FeCl$_3$. When single mutant strains *fumA*, *fumB*, and *fumC* were grown in iron-limited minimal medium, as expected, only the *fumC* mutant strain possessed a lag in growth (Fig 6A). With addition of iron in the medium the growth of each strain was improved compared to growth in iron-free conditions; however, the *fumC* mutant still possessed a lag in growth during iron conditions yet was able to reach a final OD saturation similar to wild-type (Fig 6A). Both single mutant strains *fumA* and *fumB* had no defects in either minimal media condition (Fig 6A). Therefore, iron can partially restore the growth defect of the *fumC* mutant strain to wild-type levels suggesting that iron-limitation is the basis for the required presence of FumC for optimal growth. Consistent with this notion, the double mutant strain *fumAB*, which has a functional FumC that does not require iron, grew similarly to wild-type in both conditions (Fig 6B). Double mutants lacking FumC (*fumAC* and *fumBC*) resulted a slight lag during exponential growth in iron-free minimal medium and when iron was provided (Fig 6B). The triple mutant *fumABC* was completely crippled in growth during all tested conditions, even with iron present (Fig 6B). In medium without iron, single mutant *sdhB* and double mutant *sdhBfrdA* grew slightly better than the wild-type; however, when iron was added to the medium, these mutants displayed a slight lag in growth rate compared to wild-type CFT073 and single mutant *frdA* (Fig 6C). These data may be explained by previous reports referenced above, which demonstrated the down-regulation of anaerobic respiratory genes and the up-regulation of aerobic respiratory genes during iron-limitation.

To confirm that FumA is functioning as the preferred fumarase enzyme, qPCR analysis of wild-type CFT073 and the single fumarase mutants were examined during iron-replete and iron-deplete conditions under aerobic growth. As expected, in wild-type CFT073 *fumA* was upregulated while *fumC* was downregulated during iron-replete compared to iron-deplete conditions (S3 Fig) indicating that FumA is the main enzyme in growth conditions with iron present. In fact, *fumA* expression was trending up in every mutant strain and likewise, *fumC* was downregulated (S3 Fig). These findings confirm that in aerobic growth conditions with iron available, FumA is the preferred enzyme for converting fumarate to malate in the oxidative TCA cycle, while FumC is preferred under iron-limitation.

## Disruption of SDH and the loss of FumC results in the excretion of acidic end products

To determine if the oxidative and reductive TCA pathway mutant strains behave as expected during aerobic and anaerobic conditions, phenol red in LB-1.5% agar was used as a pH indicator to test for the excretion of acidic fermentation end products. In this phenotypic assessment a change in pH, which is dependent on proton concentration, is indicated by a color change of

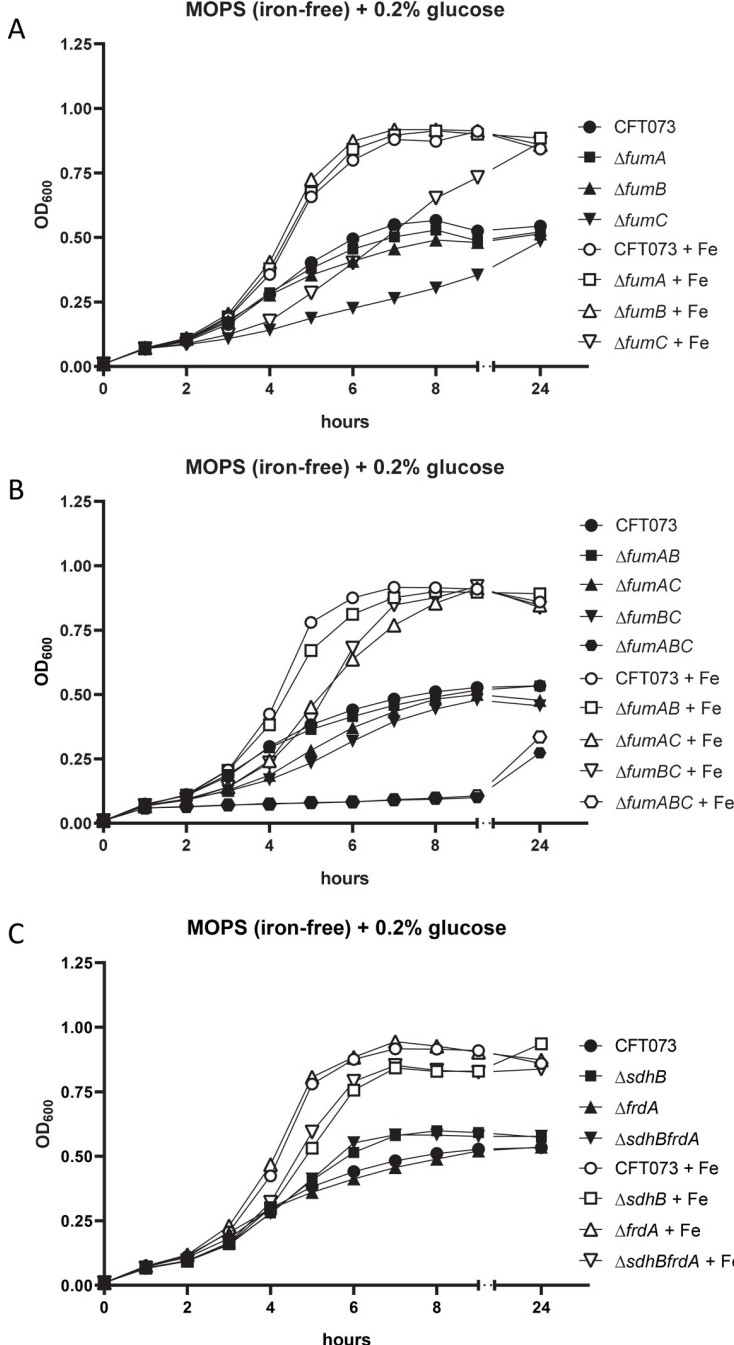

**Fig 6. The growth defect of *fumC* can be rescued with addition of exogenous iron.** Wild-type CFT073 and (A) single fumarase mutant strains *fumA*, *fumB*, and *fumC*, (B) double and triple fumarase mutant strains *fumAB*, *fumAC*, *fumBC*, *fumABC* and (C) SDH and FRD mutant strains were examined in iron-free defined medium containing 0.2% glucose with and without 36 μM FeCl$_3$. OD$_{600}$ was recorded every hour and symbols indicate the mean values of three trials. Black symbols indicate the absence of iron and white symbols denote medium containing iron.

the phenol red. Neutral pH on phenol red agar is orange, alkaline pH on phenol red agar is pink, and acidic pH on phenol red agar is yellow. Under anaerobic and aerobic conditions, the oxidative and reductive TCA pathway mutant strains were examined on plain phenol red agar, phenol red agar containing 0.2% glucose or 0.2% glycerol to mimic fermentative conditions

(during anaerobic conditions), and phenol red agar containing 0.2% glucose or 0.2% glycerol with the addition of 0.2% fumarate to serve as a final electron acceptor for anaerobic respiration (during anaerobic conditions).

Following incubation under anaerobic conditions at 37˚C on phenol red agar alone, wild-type CFT073 and all of the fumarase mutants turned orange indicating a neutral pH (Fig 7A). During anaerobic conditions on phenol red agar containing glucose or glycerol as the sole carbon source, wild-type CFT073 and the fumarase mutants were observed to be yellow indicating an acidic pH, suggesting that these strains are capable of carrying out fermentation. Exogenous fumarate in the phenol red agar containing either glucose or glycerol also resulted in a yellow color for wild-type and all of the mutant strains (Fig 7A). Interestingly, transitioning the phenol red agar plates to aerobic conditions at 25˚C, resulted in only the fumarase mutants with disruptions in *fumC* (mutant strains *fumC, fumAC, fumBC, fumABC*) to remain yellow and acidic on the phenol red agar containing glucose alone or glucose with the addition of exogenous fumarate (Fig 7A). Wild-type CFT073, single fumarase mutant strains, *fumA* and *fumB*, as well as double mutant strain *fumAB* became alkaline presenting as a pink color on the phenol red agar for all conditions(Fig 7A). Surprisingly, only the triple mutant strain *fumABC* turned yellow following aerobic incubation on phenol red agar containing glycerol alone or phenol red agar containing glycerol and exogenous fumarate (Fig 7A).

Interestingly, when the strains were examined on plain phenol red agar under aerobic conditions at 37˚C following a 6 h incubation, only the triple mutant strain *fumABC* was orange compared to wild-type CFT073 and the other single and double fumarase mutant strains which were pink (Fig 7B). Under aerobic conditions over the span of 6–48 h we noted that all of the phenol red agar conditions tested (Fig 7B) had similar results to the phenol red agar plates examined during anaerobic conditions and transitioned to aerobic conditions (Fig 7A). Fumarase mutant stains deficient in *fumC* maintained acidity on phenol red agar containing glucose and glucose with the addition of fumarate (Fig 7A and 7B). These conditions with glucose also resulted in the fumarase mutant strains deficient in *fumC* having reduced growth compared to wild-type CFT073 (Fig 7B). On phenol red agar containing glycerol or glycerol with the addition of fumarate, only the triple mutant strain *fumABC* remained acidic (Fig 7A and 7B).

Following anaerobic incubation at 37˚C, wild-type CFT073 and the SDH and FRD mutants displayed an orange hue on phenol red agar alone, while the addition of glucose or glycerol to the phenol red agar caused the strains to turn yellow (Fig 8A). Addition of fumarate as a final electron acceptor to the phenol red agar containing glucose or glycerol also resulted in a yellow color to be observed (glucose may result in a more intense yellow color than glycerol) (Fig 8A). After transitioning the phenol red agar plates to aerobic conditions for 24 h at 25˚C, all of the strains turned pink on phenol red agar alone as expected. Interestingly, single mutant strain *sdhB* and double mutant strain *sdhBfrdA* turned yellow on phenol red agar containing either carbon source with or without the addition of fumarate while wild-type CFT073 and the *frdA* mutant strain turned orange on phenol red agar containing carbon source alone to slight pink on phenol red agar with addition of a carbon source and fumarate (Fig 8A). These observations are not surprising since the *sdhB* and *sdhBfrdA* mutant strains are unable to operate an oxidative TCA cycle to carry out oxidative respiration, thus use of the exogenous fumarate as an electron acceptor during anaerobic conditions would result in a yellow color as was observed.

Following incubation of inoculated phenol red agar plates under aerobic conditions over the span of 6–48 h (Fig 8B) we noted that the strains looked similar to those inoculated on the phenol red agar plates examined during anaerobic conditions transitioned to aerobic conditions (Fig 8A). We also observed that the *sdhB* and *sdhBfrdA* mutant strains did not grow as well as wild-type CFT073 and the *frdA* mutant strain (Fig 8B). SDH mutant strains maintained acidity on phenol red agar containing carbon sources, glucose or glycerol, and also on phenol

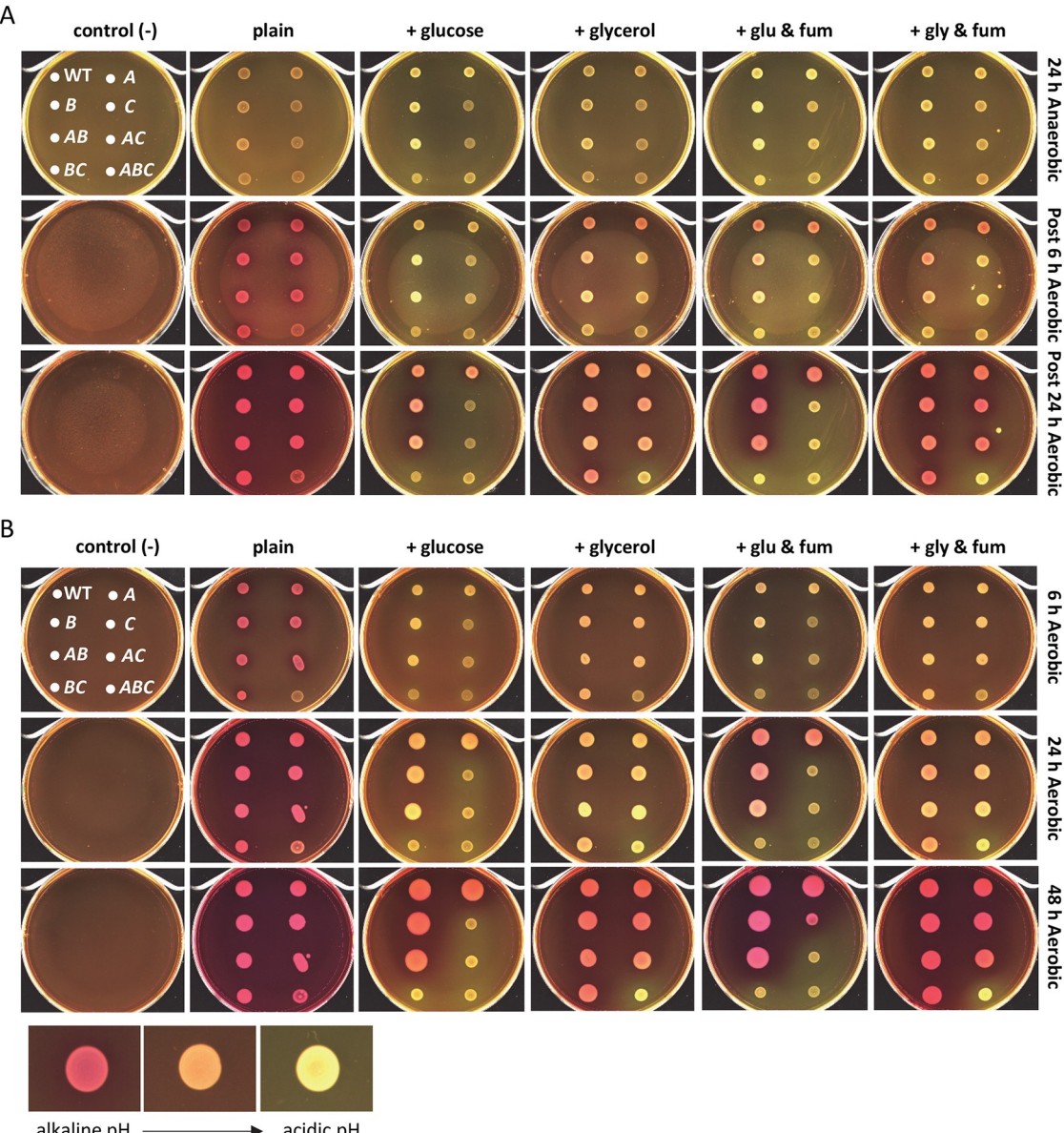

**Fig 7. Loss of FumC results in acidic end products.** Wild-type CFT073, single, double, and triple fumarase mutants were examined on LB agar containing phenol red (plain), LB agar containing phenol red and 0.2% glucose, LB agar containing phenol red and 0.2% glycerol, LB agar containing phenol red, 0.2% glucose, and 0.2% fumarate, and LB agar containing phenol red, 0.2% glycerol, and 0.2% fumarate. Un-inoculated phenol red agar plates with orientation of strains are located on the right and were treated under the same conditions. Alkaline pH is pink, neutral pH is orange, and acidic pH is yellow. (A) Plates were spotted with bacterial culture, incubated under anaerobic conditions at 37°C for 24 h, imaged, and then transitioned to aerobic conditions at 25°C and imaged at 6 h and 24 h. (B) Plates were spotted with bacterial culture, incubated under aerobic conditions at 37°C, imaged at 6 and 24 h, transitioned to 25°C and imaged at 48 h.

red agar containing a carbon source with the addition of fumarate (Fig 8A and 8B). These findings suggest that disruptions within enzymes, FumABC and SdhBFrdA that function in both the oxidative and reductive TCA pathways cause acidic end products to be excreted under anaerobic as well as aerobic conditions. Loss of oxidative TCA cycle enzyme SdhB alone also results in acidic end products following anaerobic and aerobic conditions.

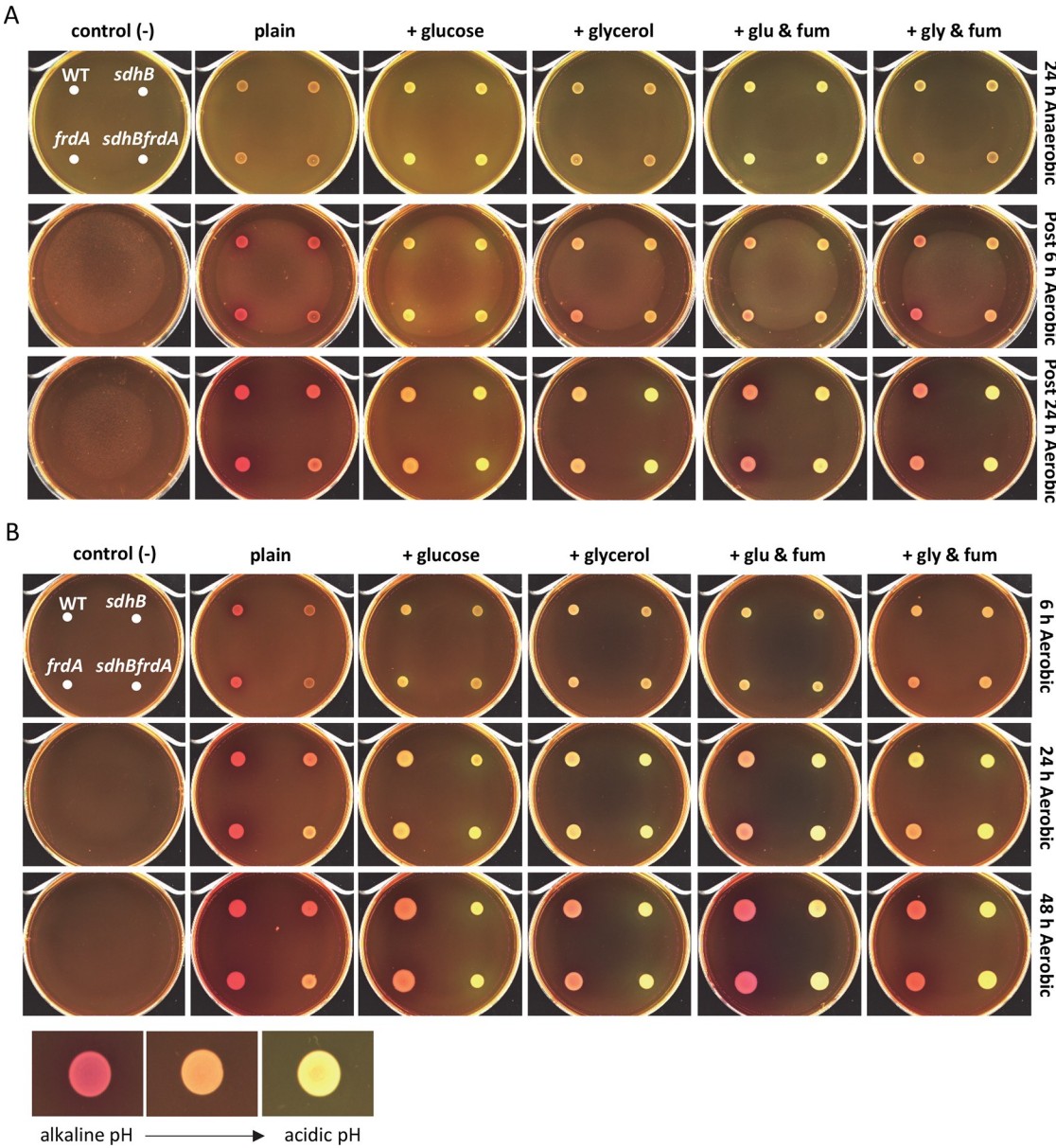

**Fig 8. Disruption of SDH results in acidic end products.** Wild-type CFT073, single mutant strains, *sdhB* and *frdA*, and double mutant strain *sdhBfrdA* were examined on LB agar containing phenol red (plain), LB agar containing phenol red and 0.2% glucose, LB agar containing phenol red and 0.2% glycerol, LB agar containing phenol red, 0.2% glucose, and 0.2% fumarate, and LB agar containing phenol red, 0.2% glycerol, and 0.2% fumarate. Un-inoculated phenol red agar plates with orientation of strains are located on the right and treated under the same conditions. Alkaline pH is pink, neutral pH is orange, and acidic pH is yellow. (A) Plates were spotted with bacterial culture, incubated under anaerobic conditions at 37˚C for 24 h, imaged, and then transitioned to aerobic conditions at 25˚C and imaged at 6 h and 24 h. (B) Plates were spotted with bacterial culture, incubated under aerobic conditions at 37˚C, imaged at 6 h and 24 h, transitioned to 25˚C and imaged at 48 h.

## Disruption of both the oxidative and reductive TCA pathway results in sensitivity to oxidative and acid stress

During respiration the reduced electron carriers NADH and $FADH_2$ formed during the oxidative reactions of the TCA cycle ultimately transfer electrons to oxygen as a final electron acceptor providing a proton gradient for the cell to produce energy by oxidative phosphorylation. Despite this aerobic respiratory chain being highly efficient in suppling energy and generating

proton motive force, damaging reactive oxygen species, such as superoxide ($O_2^-$), hydrogen peroxide ($H_2O_2$), and hydroxyl radicals, can also be produced resulting in oxidative damage to the cell. These reactive oxygen species oxidize and inactivate [4Fe-4S] cluster-containing dehydratases such as FumA and FumB [28, 31–33]. As a countermeasure against oxidative stress, cellular defense mechanisms including superoxide dismutases (SODs) help convert the hydroxyl radicals to $O_2$ and $H_2O_2$ which in turn induces the oxidative stress regulator, OxyR, to activate expression of a cohort of scavenging and repair enzymes. FumC is a known member of the superoxide response (*soxRS*) regulon [34] and lacks an iron-sulfur cluster making it oxygen-stable and unaffected by oxidative stress compared to its labile FumA and FumB counterparts.

We reasoned that the viability of the oxidative and reductive TCA pathway mutant strains would be decreased during oxidative stress due to their presumed partial defects in cellular respiration. To determine a suitable bactericidal concentration of $H_2O_2$ with at least three logs of killing of wild-type CFT073, various concentrations of $H_2O_2$ were examined over a 60-minute time period. For this pilot experiment wild-type CFT073 was inoculated into LB medium containing 0.0%, 0.03%, 0.3% and 3% $H_2O_2$ and sample was removed every 10 minutes for 1 hour, diluted, and drip-plated on plain agar to determine colony forming units (CFU). We observed optimal killing of wild-type in LB medium containing 0.3% $H_2O_2$ (S4A Fig), defined by a 3-log decrease in CFU, and examined the bactericidal effects of $H_2O_2$ at this concentration for all of the oxidative and reductive TCA pathway mutants. To control for any potential growth of the mutants during this assay, we incubated all TCA pathway mutants in LB medium containing 0% $H_2O_2$ which resulted in no statistically significant difference in CFU between wild-type CFT073 during a 30-minute incubation (S4B Fig). Triple mutant strain *fumABC* was observed to have the greatest susceptibility to 0.3% $H_2O_2$ at 20, 30, and 40 minutes post-incubation with a statistically significant decrease in CFU compared to wild-type CFT073 (*P*< 0.0001) (Fig 9A). This trend of susceptibility to 0.3% $H_2O_2$ by the *fumABC* triple mutant was observed until 40 minutes post-incubation at which time detectible viability was eliminated (Fig 9A). Here we see that complete loss of both the oxidative and reductive TCA pathways in which the reversible reaction between fumarate and malate (*fumABC*) was disrupted results in an increased sensitivity to *in vitro*-induced exogenous oxidative stress. At 40 minutes post-incubation, the *sdhB* mutant was also found to have a significantly decreased CFU compared to CFT073 (*P* = 0.0003) (Fig 9A). All mutant strains tested lost viability 10 minutes earlier than the wild-type, indicating that intact cellular respiration capabilities improve overall survival to exogenous sources of oxidative stress. Our inability to observe minor differences of killing between the mutant strains may be due to the lack of this assay's sensitivity to distinguish the susceptibility of the mutants to $H_2O_2$ during this assay.

During acid stress, the bacterial cell undergoes physiological, metabolic, as well as proton-consuming acid resistance mechanisms that reduces influx of protons into the cell. Under normal growth conditions, proton motive force is generated by the coupling of metabolic redox reactions with the export of protons. However, during aerobic growth under mild acid stress (pH 5.0–7.0) electron transport chain components including cytochrome oxidase (*cy*o) genes, NADH dehydrogenase II (*ndh*) genes, succinate dehydrogenase (*sdh*), and NADH dehydrogenase I (*nu*o) genes have been found to be upregulated [35, 36]. This enables the cell to counteract intracellular increases in pH by exporting protons. To test the viability of TCA cycle mutant strains in acidic conditions, we cultured bacteria in LB buffered to pH 7, pH 5, and pH 2.5 for one hour before enumerating CFU. At pH 7 and pH 5, none of the mutants displayed significantly altered CFU compared to wild-type CFT073 (S5A and S5B Fig, respectively). However, when grown at pH 2.5, the triple *fumABC* mutant had significantly reduced CFU (*P*<0.0001), which indicates an increased susceptibility to acid-induced stress (Fig 9B). The

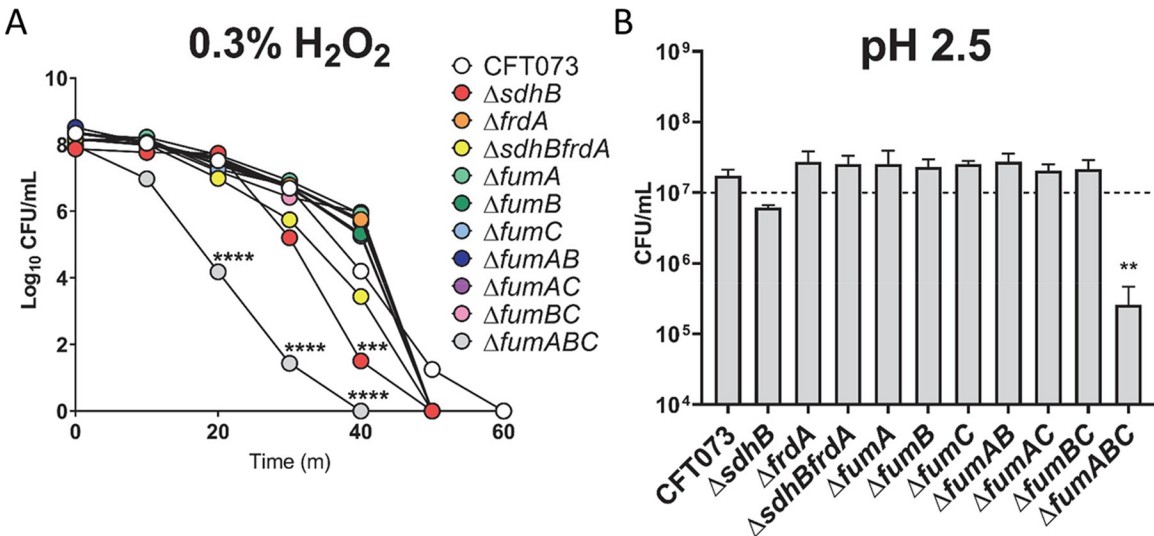

**Fig 9. Disruption of both the oxidative and reductive TCA pathway results in sensitivity to oxidative stress.** (A) Wild-type CFT073 and the oxidative and reductive TCA pathway mutant strains were examined in LB medium containing 0.3% $H_2O_2$. Samples were collected every 10 minutes, diluted, and plated on LB agar to determine CFU. Colored symbols indicate the mean of three independent biological replicates. Two-way ANOVA was use to assess statistical differences at each time point (***$P<0.001$, ****$P<0.0001$). (B) Wild-type CFT073 and the oxidative and reductive TCA pathway mutant strains were incubated in 100 mM MES buffered LB, pH 2.5, for 60 minutes, diluted, and plated on LB agar to determine CFU. The dashed line represents the intended input CFU/ml at time zero. The mean is indicated by the bar height and the error bars display SEM of three biological replicates.

*sdhB* mutant also had reduced CFU, although this was not statistically significant (Fig 9B). Collectively, these *in vitro* experiments suggest that the *fumABC* and *sdhB* mutants are more susceptible to exogenous radicals and acid stress.

Growth in a variety of media, including in human urine, was not sufficient to explain the negative impact of *fumC* mutations during UTI. Similarly, it was not possible to invoke growth defects to explain the conditional and detrimental impact of FRD on UPEC fitness. Therefore, we sought to determine if the increased susceptibility to radicals and acidic pH in TCA cycle mutant strains might provide explanation for their observed fitness defects *in vivo*. Because these *in vitro* experiments partially mimic bactericidal mechanisms deployed by neutrophils, we conducted pharmacological neutrophil depletion experiments in mice prior to performing co-challenge. If this is indeed the underling mechanism contributing to fitness defects, we predicted that neutrophil depletion using cyclophosphamide would rescue the *in vivo* phenotypes observed for the mutant strains *fumC* and *sdhB*. While blood tests indicated that the pharmacological depletion of neutrophils was successful and we observed a 3–4 log increase in total CFUs comparing cyclophosphamide- to PBS-treated mice (S6A and S6C Fig), the observed fitness defects in *fumC* and *sdhB* were not reversed by neutrophil depletion (S6B and S6D Fig). Howeverr, the analysis of blood samples did indicate that in addition to neutropenia, the mice were also anemic. It remains possible that the anemia is contributing to the increased fitness defect ($P = 0.0101$) observed for the *fumC* mutant in the bladders of cyclophosphamide-treated mice (S6B Fig). These results further support the notion that iron availability *in vivo* could be a substantial contributing factor to the preferential utilization of FumC in the urinary tract.

## Absence of FumC facilitates resistance to bactericidal antibiotics

Currently, the treatment for UTI is a course of antibiotics, most commonly trimethoprim or ciprofloxacin for more serious cases. These two antibiotics have different methods for

inhibiting infection, by deploying bacteriostatic or bactericidal mechanisms. Bacteriostatic antibiotics prevent bacterial growth without affecting viability, and in the clinical setting, would allow the host immune system to clear the infection. In contrast, bactericidal antibiotics effectively eliminate the viability of bacteria and have even been proposed to produce these lethal effects in part by hyperactivation of the respiratory chain and concomitant production of reactive species [37]. To better understand the role for the TCA cycle in *E. coli* susceptibility to clinically relevant antibiotics, all mutants and wild-type CFT073 were cultured in LB medium containing a single antibiotic. Three bacteriostatic antibiotics were chosen for these experiments: chloramphenicol, trimethoprim, and tetracycline. Strains were inoculated at ~$10^7$ CFU and incubated in 10 μg/ml chloramphenicol for 180 minutes. The growth of all strains was similarly stunted by the presence of chloramphenicol (Fig 10A). In 5 μg/ml of tri-methoprim, *sdhBfrdA* was the only strain differentially affected; however, the decrease in CFU was minimal and not significant (Fig 10B). CFU of the triple *fumABC* mutant strain was decreased by 2.5-logs over the 180-minute incubation with 10 μg/ml tetracycline (Fig 10C). Although not statistically significant, this trend may indicate that absence of all fumarase activity is an important compensation mechanism when protein translation is inhibited.

Bactericidal antibiotics are effective at killing bacteria by inhibiting essential cellular mechanisms such as cell wall synthesis, protein translation, and DNA replication. We performed two *in vitro* methods; a liquid LB medium assay containing antibiotic and a Mueller-Hinton agar-based assay using antibiotic test strips to examine the bactericidal effects of ciprofloxacin, ampicillin, and streptomycin on the respiration mutants. Bacteria cultured with 250 ng/ml cip-rofloxacin in LB medium for 3 hours displayed up to 4-logs loss of CFU (S7A Fig). Both the *frdA* and *fumAC* mutant strains were more resistant to ciprofloxacin than wild-type CFT073, displaying no decrease in CFU over time (S7A Fig). The *sdhB* and *sdhBfrdA* mutant strains both had a 2-log decrease in CFU within 45 minutes, indicating increased susceptibility to cip-rofloxacin. The minimum bactericidal concentration (MBC) was determined to be 0.047 μg/ml for the triple mutant strain *fumABC*, compared to 0.010 μg/ml for wild-type CFT073 (Fig 10D). This suggests that loss of fumarase activity increases tolerance to bactericidal antibiotics targeting DNA gyrase. When incubated in LB medium with 5 μg/ml ampicillin, every mutant strain displayed a higher final CFU compared to CFT073 (S7B Fig). The CFU of mutant strains *fumABC* and *fumBC* increased in the presence of ampicillin, demonstrating no effect of the antibiotic during 3 hours of incubation, unlike wild-type CFT073 which decreased by 2.5-logs (S7B Fig). Similarly, results from agar plates with an antibiotic test strip indicated that the *fumAC*, *fumBC*, and *fumAB* mutant strains were the least susceptible to ampicillin (Fig 10E). However, unlike the results obtained from LB medium containing ampicillin, mutant strains *sdhB* and *sdhBfrdA* had a lower MBC than wild-type CFT073 (Figs 10E and S7B). The most striking bactericidal effect was with the use of streptomycin, which inhibits protein synthesis. In liquid culture, the *sdhB* and *sdhBfrdA* mutant strains had a 5.5-log decrease in CFU, wild-type had a 4-log decrease in CFU, and the triple mutant *fumABC* displayed an increase in CFU over time (S7C Fig). Interestingly, the four most resistant strains to 5 μg/ml streptomycin in the liquid assay were those lacking FumC. These results were consistent with the findings that mutants lacking *fumC* examined on the Mueller-Hinton agar with a streptomycin test strip demonstrated decreased susceptibility (Fig 10F). These findings are consistent with the notion that certain bactericidal antibiotics depend upon either a robust electrochemical gradient across the cell membrane, generated during maximal re-oxidation of reduced NADH to NAD$^+$, or even that bactericidal antibiotics cause hyper-activation of membrane respiration and formation of reactive oxygen species within the cell [37] since many of the tested TCA cycle mutants are more resistant to these bactericidal antibiotics than the parental strain.

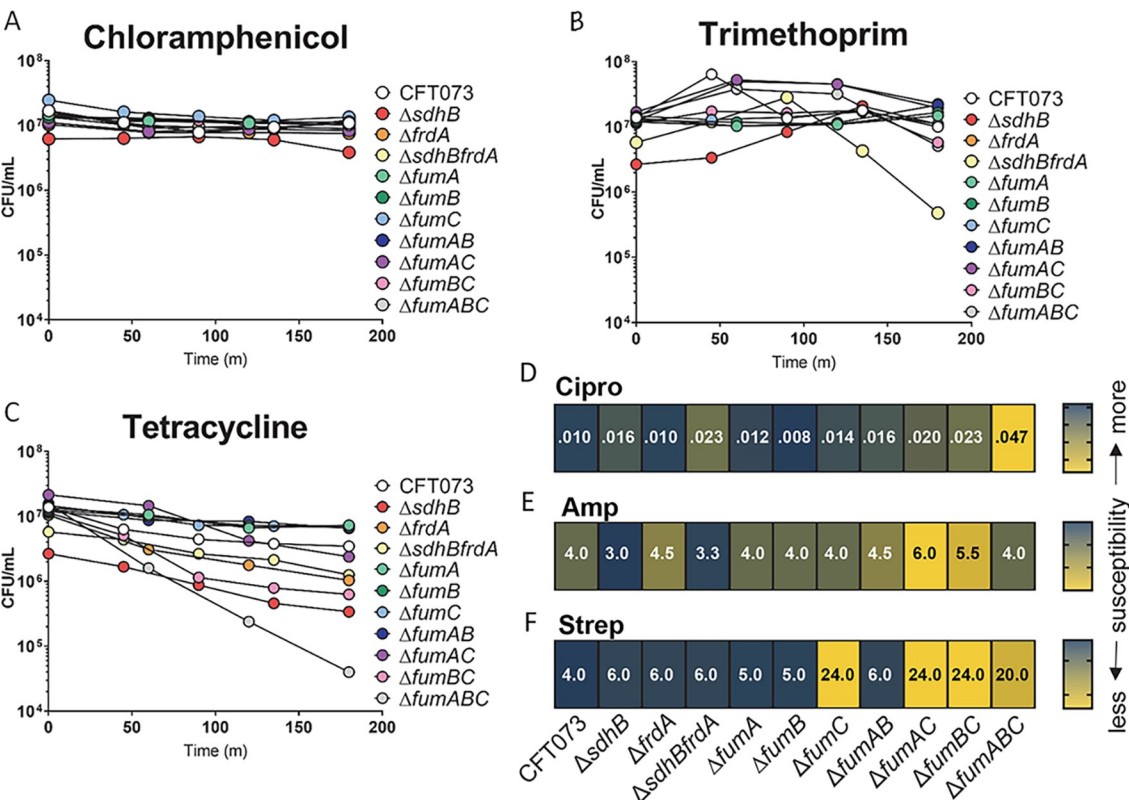

**Fig 10. Absence of FumC leads to decreased susceptibility to bactericidal antibiotics.** Wild-type CFT073 and the oxidative and reductive TCA pathway mutant strains were examined in LB medium containing (A) chloramphenicol 10 μg/ml (B) trimethoprim 5 μg/ml or (C) tetracycline 10 μg/ml. Samples were collected over the course of 180 minutes, diluted and plated on LB agar to determine CFU. Colored dots signify the mean of three biological replicates. The Minimum Bacterial Concentration (MBC) of (D) ciprofloxacin 0.002–32 μg/ml (E) ampicillin 0.016–256 μg/ml or (F) streptomycin 0.064–24 μg/ml observed for wild-type CFT073 and the mutant strains were recorded following overnight incubation on Mueller-Hinton agar with antibiotic test strips. The values shown are averages from two independent trials.

To better understand these results, we performed identical MBC determinations under anaerobic conditions. We observed a general trend toward reduced susceptibility to bactericidal antibiotics across all of the strains tested (S8 Fig). The results for ampicillin and ciprofloxacin under anaerobiosis were similar to what was observed under aerobic conditions, where double and triple mutants containing a disruption in *fumC* were 2- or 3-fold more resistant than the parental strain (S8A and S8B Fig). As expected, we observed a dramatic increase in resistance to the aminoglycoside streptomycin under anaerobic conditions where the parental CFT073 MBC shifted over 10-fold from 4 μg/ml to 48 μg/ml (Figs 10F and S8C). This indicates that the decreased PMF that occurs during anaerobic growth likely results in decreased uptake of the aminoglycoside and reduced susceptibility, which was expected because it is known that aminoglycosides are ineffective toward anaerobic microbes. However, similar to what was observed under aerobic conditions, any strain lacking FumC was more resistant to streptomycin than the parental strain; *fumC* 79 μg/ml, *fumAC* 128 μg/ml, *fumBC* 112 μg/ml, and *fumABC* 128 μg/ml (S8C Fig). While far from conclusive, the collective results from the MBC experiments demonstrate some unappreciated linkage between the TCA cycle and the mechanism underlying susceptibility to these bactericidal antibiotics.

## *E. coli* motility is dependent on FumC

Bacterial swimming motility is an easily quantified phenotype that can be used to indirectly assess the electrochemical gradient across the cell membrane. Flagella rotation is dependent on proton motive force (PMF), or specifically the $H^+$ gradient created by membrane respiration during the oxidation of the reduced electron carrier NADH produced by the oxidative TCA cycle. During this process electrons are passed through an electron transport chain and protons are deposited across the cytoplasmic membrane. The movement of $H^+$ down the chemiosmotic gradient, through the flagellum stator drives the rotation of flagella. Because flagella-mediated motility is required for colonization of the urinary tract and contributes to UPEC virulence [38] we sought to determine if the fitness defects of the oxidative and reductive TCA pathway mutant strains during UTI was due to a deficiency in motility. To address this, mutant strains were spotted onto 0.25% soft agar and their swimming phenotypes were compared to that of wild-type CFT073. Only double mutant strain *sdhBfrdA* had a hypermotile swimming phenotype with a diameter of 58.75 mm compared to the swimming diameter, 36.42 mm, of wild-type CFT073 (*P* = 0.0001) (Fig 11A and 11B). Interestingly, only mutants with a disruption within one of the fumarase encoding genes, *fumC*, had a hypomotile swimming phenotype compared to the swimming phenotype of wild-type CFT073. Single mutant strain *fumC*, double mutant strains *fumAC* and *fumBC*, and triple mutant strain *fumABC* were significantly decreased with swimming diameters of 7.53 mm, 6.59 mm, 8.25 mm, and 7.61 mm, respectively (*P*<0.0001) (Fig 11A and 11B). These findings suggest that the *in vivo* fitness defects observed for the oxidative and reductive TCA pathway mutant strains with disruptions in FumC may be due to impaired motility influencing poor UPEC colonization within the urinary tract.

It has been previously reported that flagellar expression and motility is directly affected by the overexpression of type 1 fimbriae (*fim*) whose promoter is located within an invertible element (IE) that is capable of switching from an on to an off orientation and vice versa [39, 40]. While decreased motility and flagellar expression can result from constitutive expression of the *fim* operon, increased motility is not indicative of decreased *fim* expression [39]. We wanted to better understand if this reciprocal relationship could explain the motility findings for strains lacking *fumC* or for the *sdhBfrdA* double mutant. We found that the *fim* promoter was in the OFF-orientation for 92%-100% of the bacterial population for the majority of the respiration mutant strains; single mutant strains *sdhB* and *fumC*, double mutant strains *sdhBfrdA*, *fumAB*, *fumAC*, *fumBC*, and triple mutant strain *fumABC* (Figs 11C and S9A). These findings suggest that the decreased motility of the single, double, and triple mutant strains with a disruption in *fumC* is not due to overexpression of *fim* since the IE of 92%-100% of the bacterial population was in the OFF-orientation. Likewise, the hypermotile swimming phenotype of double mutant strain *sdhBfrdA* cannot be explained by the orientation of the IE since 97% of the bacterial population was in the OFF-orientation. There is a moderate relationship ($R^2$ = 0.87) between motility and the percent of bacteria population with the *fim* promoter in the ON-orientation within the fumarase mutant strains; however, this correlation is the inverse of expected results (S9B Fig). Interestingly, there is also a moderate relationship ($R^2$ = 0.70) between motility and the percent of bacteria population with the fim promoter in the ON-orientation within the SDH and FDR mutant strains (S9C Fig).

To determine if transcription is responsible for the motility phenotypes of the oxidative and reductive TCA pathway mutant strains, quantitative PCR was performed to quantify the transcription of flagellar and fimbrial genes (S10 Fig). Expression of the RNA polymerase sigma factor for the flagellar operon, *fliA*, and flagellin, *fliC*, was downregulated greater than 2-fold in the single mutant strain *fumC*, double mutant strains *fumAB*, *fumAC*, *fumBC*, and the triple

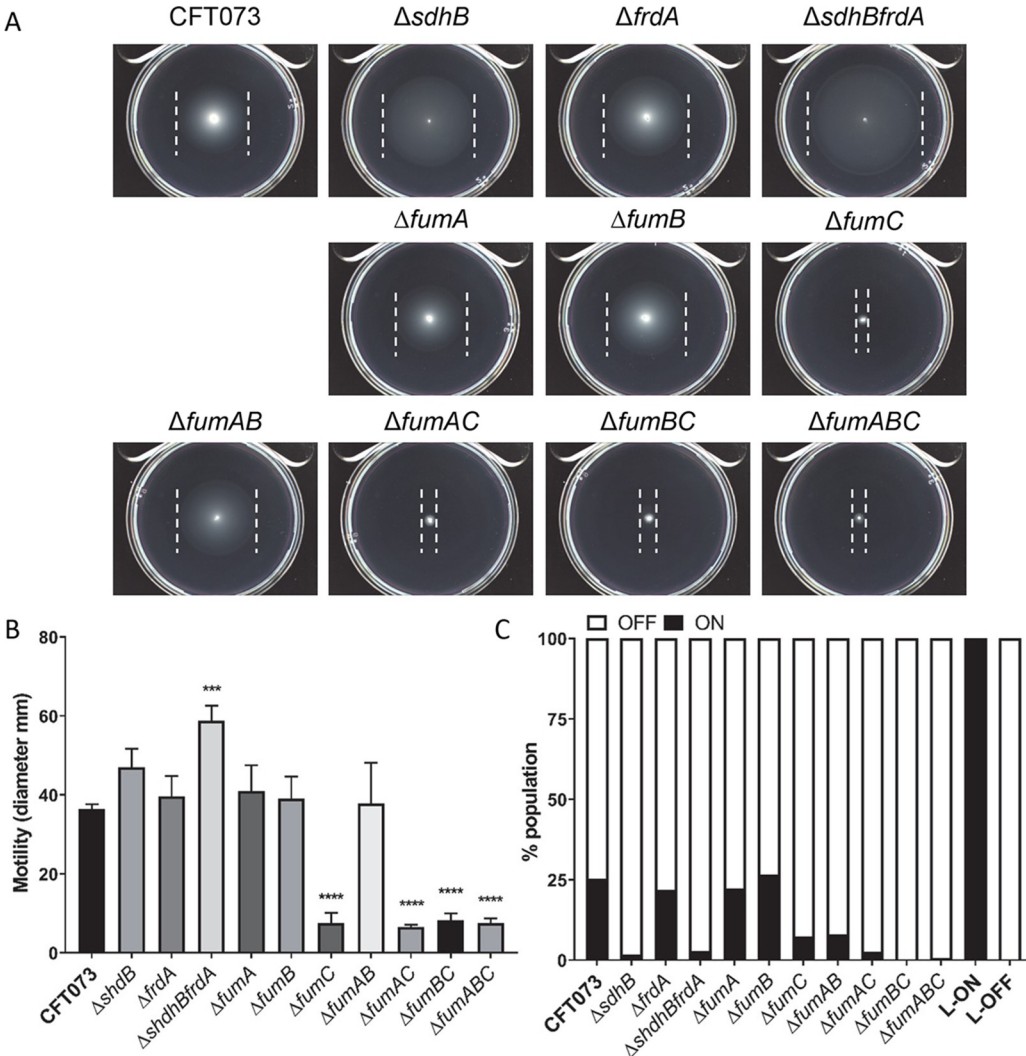

**Fig 11. FumC is required for motility in CFT073.** (A) Wild-type CFT073 and each oxidative and reductive TCA pathway mutant strain was stabbed into semi-soft motility agar and incubated overnight at 30°C for 16 h. Images of representative motility agar plates following incubation with inoculated strains. Outer swim edge is indicated by a white line. (B) Bars represent the average swimming diameters (in mm) of biological triplicates. Error bars represent the standard errors of the means (SEM). Statistical significance was determined by one-way ANOVA (***$P$ <0.001, ****$P$ <0.0001). (C) The invertible element (IE) assay was performed on strains grown statically in LB at 37°C for 18 h and standardized to $OD_{600}$ = 0.5. Bars depict percent population of bacteria with the *fim* promoter within the IE positioned in the on orientation (black bar) or off orientation (white bar) as calculated with pixel density values for bands of PCR products in (Fig 9A).

mutant strain *fumABC* compared to wild-type CFT073 (S10A and S10B Fig). Gene expression for the pyelonephritis associated pili subunits, *papA* and *papA2* and type one fimbriae major subunit, *fimA*, was also downregulated greater than 2-fold, although not statistically significant, in the single mutant *fumC*, double mutant strains *fumAB*, *fumAC*, *fumBC*, and the triple mutant strain *fumABC* compared to wild-type CFT073 (S10C, S10D and S10E Fig). As expected, expression of *fimA* for the CFT073 *fim* L-ON strain, in which the type 1 *fim* promoter is mutated in the on-orientation, and the CFT073 *fim* L-OFF strain, in which the type 1 *fim* promoter is mutated in the off-orientation was upregulated and downregulated, respectively, greater than 5-fold compared to wild-type CFT073 (S10E Fig). Up-regulation or down-

regulation of flagellar and fimbrial gene expression in the SDH and FRD mutant strains was not observed (S10 Fig). These transcriptional findings suggest that it is likely a physiological mechanism, for example disruptions in the TCA affecting the proton gradient across the membrane, playing a role in the reduction of motility in the absence of *fumC* and the increase in motility in the absence of SDH and FRD.

## Discussion

The urinary tract is a moderately oxygenated environment in which *E. coli* replicates using flexible metabolic pathways and by deploying key virulence factors. As facultative bacteria, UPEC can adapt to aerobic and anaerobic environments by substituting and altering available respiratory components within a vast modulatory network. Because of this intricate respiratory system, it is plausible that subset populations of UPEC occupying various niches within the urinary tract are capable of differentially expressing these respiratory pathways as a means to satisfy the metabolic requirements of that microenvironment [12]. Our recent studies on *E. coli* metabolism during extraintestinal colonization indicate that gluconeogenesis and the TCA cycle as well as peptide import are required for infection [1, 5, 8]. While our previous work identifies FumC of the oxidative TCA cycle as required during UPEC infection [5], it is known that the *E. coli* genome encodes two additional fumarase enzymes. Interconversion of malate to fumarate is generally catalyzed by one of three fumarases; FumC and FumA catalyzes fumarate oxidation to malate in the oxidative TCA cycle and FumB catalyzes malate reduction to fumarate in the branched, reductive TCA pathway. The present study provides a more complete assessment for the role of all three known fumarases during UTI while also examining succinate dehydratase (SDH) and fumarate reductase (FRD) which interconvert fumarate and succinate during the complete oxidative and branched reductive TCA cycle or pathway, respectively. In this study we further investigated the requirements these TCA pathways during UPEC infection and performed *in vitro* iron-limitation, oxidative stress, acid resistance, antibiotic susceptibility, and motility assays to assess the roles of these respiratory enzymes under a variety of conditions thought to mimic the urinary tract environment and associated host defense mechanisms.

Here we report that oxidative TCA cycle enzyme FumC has the largest contribution to UPEC colonization in the bladder and kidneys of the iron-limited urinary tract, while FumA and FumB are dispensable *in vivo*. This is in contrast to a *serovar* Typhimurium SR-11 *fumB* mutant found to be avirulent within the intestine [24]. Additionally, a *fumAC* mutant in the same strain was attenuated, indicating that either FumA or FumC or both play a role during infection [24]. Utilizing the UPEC double mutants (*fumAB*, *fumAC*, *fumBC*), we determined that FumA cannot replace FumC during UTI. In fact, the *fumAB* strain had a fitness advantage *in vivo*, suggesting that loss of FumA could be beneficial during infection. This may be due to the energy costs associated with the production or activity of the enzyme and its requirement for coordinating $Fe^{3+}$ in an iron-limited environment. It is notable that FumC lacks an iron-sulfur cluster and its activity is not iron-dependent, unlike FumA and FumB. Because the urinary tract is iron-limited, it is not surprising that loss of *fumC* in UPEC results in a fitness defect. Furthermore, reduced growth of the *fumC* mutant strain during *in vitro* iron-limited conditions supports that FumA and FumB both require iron for their activity. Our data strongly suggests that the oxidative fumarase reaction is important for UPEC during UTI and specifically FumC is the most efficient enzyme for converting fumarate to malate in this system.

Our previous studies also indicated that SDH is required for *in vivo* colonization of the urinary tract while FRD is dispensable [1, 5]. These findings are dissimilar from studies in which

inactivation of fumarate reductase, *frdA*, reduced *E. coli* colonization in the mouse intestine [13]. However, deletion of the FRD operon in *serovar Typhimurium* strain SR-11 resulted in a fully virulent strain while a SDH operon mutant strain was attenuated following oral infection of mice suggesting that the branched TCA pathway is not important for infection in the gut [14]. Interestingly, here we report that loss of both the oxidative and reductive TCA cycle pathways in the *sdhBfrdA* double mutant was found to have no defect *in vivo* during UTI suggesting that the previous fitness defect observed due to the loss of SDH can be reversed presumably as a result of the fitness advantage that occurs when FRD is absent. Unlike the fumarase enzymes, SDH and FRD promiscuously interconvert succinate and fumarate independently of aerobic or anaerobic conditions; however, their catalytic activities are not with equal efficiencies [22]. It is conceivable that in the *sdhB* mutant, FRD is activated in response to succinate buildup in the cell. While this suits the bacterial cells' demand to convert succinate to fumarate independent of the loss of SDH, the catalytic activity of FRD in the SDH mutant produces endogenous superoxide *in vivo* [41, 42]. Therefore, we surmise that the *sdhB* mutant is able to carry out oxidative respiration via the aerobic TCA cycle; however, this occurs with a higher endogenous load of free radicals which could explain the observed *in vivo* defect during UTI.

It is important to note that the SDH mutant strain used in this study has a disruption in the last gene of the *sdhCDAB* operon which could render this mutant strain still capable of expressing *sdhC*, *sdhD*, and *sdhA*. It is plausible that hybrid enzyme complexes could be forming in the *sdhB* and *sdhBfrdA* mutant backgrounds. Substitution of SdhA with FrdA in a hybrid structure would convert succinate to fumarate at a ratio of 1:1.5 compared to functional SDH and would generate more endogenous radicals [21, 43]. However, the double mutant *sdhBfrdA* and the *sdhB* mutant have identical phenotypes in each of the *in vitro* experiments examined in this study, therefore; the presence of FRD activity is more detrimental during UTI rather than the loss of SDH activity. Recent studies have indicated the growth rate of UPEC to be extraordinarily high during UTI [44] this rapid growth could facilitate the generation of reactive species by respiratory components like FRD in order to meet the metabolic needs of the cell during infection.

Because flagella-mediated motility is associated with UPEC virulence and *in vivo* fitness defects were observed for strains deficient in *fumC*, we examined the *in vitro* swimming phenotype of each mutant strain. We determined that the decreased motility of mutant strains lacking *fumC* was not linked to type 1 *fim* expression, as the type 1 *fim* invertible element (IE) assay revealed all strains were in the OFF position. These findings suggest that the inhibited motility is not due to expression of the type 1 *fim* operon as historically shown [39, 40] and instead may be associated with proton motive force deficiencies across the cellular membrane. However, further analysis of additional site-specific recombinases within the CFT073 genome that have been reported to affect the phase variation of the type 1 promoter should be examined to rule out their influence on the oxidative and reductive TCA pathway mutant strains [45–47]. The motility defects observed for mutant strains lacking *fumC* may contribute to the observed *in vivo* fitness defects due to their inability to ascend and colonize the urinary tract. Similarly, the *sdhBfrdA* mutant had increased motility which may, in part, explain the *in vivo* fitness advantage. This could indicate that an alternate respiratory chain is used to maintain the proton motive force, which has been demonstrated in *P. mirabilis* swarming [48]. This theory can be further supported by the inability of *sdhB* and *sdhBfrdA* mutant strains to perform aerobic respiration on pH indicator-phenol red agar plates containing sugars. Both the single SDH mutant and the double SDH FRD mutant strain had less type 1 *fim* positioned in the ON orientation, but this was not reflected at the gene expression level. Together, these observations suggest that partial or complete inhibition of the TCA cycle affects virulence phenotypes of the

cell, not at the level of gene regulation, but by modulating the allocation of protons or other electrochemical gradients.

During UTI, exogenous stressors to UPEC including neutrophil infiltration, acidic pH, and antimicrobial peptides can hinder the survival and persistence of UPEC. Upon engulfment of a pathogen, neutrophils utilize hydrogen peroxide, defensins, and acidic pH to kill the invading bacterium. We found that *fumABC* and *sdhB* mutant strains were the most susceptible to exogenous hydrogen peroxide and acidic conditions which could be contributing factors to their respective *in vivo* defects during UTI. It was previously reported that the *fumABC* mutant has an altered membrane potential [49], which could explain the observed decreased resistance to exogenous stressors. However, when directly testing this hypothesis using pharmacological means to deplete neutrophils prior to co-challenge, we found that increased susceptibility to bactericidal mechanisms of neutrophils to be unsupported and failed to explain the observed fitness defects at least for *fumC* and *sdhB* mutant strains.

Another phenotype that has been somewhat controversially proposed to relate to reactive species, is that reactive oxygen generated via Fenton chemistry and hyperactivation of the respiratory chain underlies the killing mechanism of bactericidal antibiotics and lowering the steady-state of NADH increases resistance to bactericidal antibiotics [37, 50–52]. Therefore, we predicted disruptions in the aerobic TCA cycle would create a higher MBC for some of our mutant strains. All of the mutant strains lacking *fumC*, as well as the other oxidative mutant strains, displayed increased resistance to ciprofloxacin and streptomycin, which targets DNA gyrase and inhibits protein synthesis, respectively. Although these antibiotics have different mechanisms of action, they are both bactericidal and theoretically induce the production of reactive oxygen species within the cell [50]. Intriguingly, performing the same MBC determinations under anaerobic conditions resulted in nearly identical increased resistance for any strains lacking *fumC*. This suggests an unappreciated link between the TCA cycle and the killing mechanism behind certain bactericidal antibiotics that cannot be fully explained by reactive oxygen production due to the respiratory chain since it was also observed under anaerobic conditions. However, the centrality of *fumC* in our study does suggest that FumC is the functioning fumarase during antibiotic stress presumably due to its ability to continue oxidative reactions during oxidative stress due to the lack of an Fe-S cluster, and the absence of FumC under these conditions may cripple the TCA cycle and thus provide some degree of resistance to bactericidal antibiotics by some unknown mechanism.

Collectively, our findings demonstrate that the oxidative TCA cycle is critical for UPEC infection and virulence properties during UTI. Specifically, FumC is vital for UPEC colonization of the urinary tract. Strains lacking FumC display defects during iron-limited growth and reduced motility, which are required virulence mechanisms during infection. Although FumA is typically the preferred enzyme, the environmental conditions, presumably iron-limitation in the urinary tract, promote a shift to favor FumC reactions. Similarly, loss of SDH result in reduced *in vivo* fitness in the presence of functional FRD, increased sensitivity to reactive oxygen species and acid stress, and also impacts the orientation of the *fim* promoter. Our findings indicate that while neither SDH nor FRD are required during UTI, the presence of FRD in the absence of SDH is detrimental. Glycolytic shunts and alternative pools of intracellular succinate are potential explanations for the unnecessary enzymatic reactions typically carried out by SDH and FRD, which are not required during UTI. Having multiple fumarase enzymes and interchangeable SDH and FRD are well known examples of modularity within the TCA cycle and the respiratory chain. This metabolic flexibility is an adaptive trait of UPEC that utilizes redundant energy pathways in order to survive within multiple niches and also supports the hypothesis that modulating the electrochemical gradient across the cytoplasmic membrane could be a beneficial strategy to optimally replicate within the host environment [49]. Our

findings support the premise that UPEC shifts from utilizing glycolysis and sugar sources as a means for survival in the gut environment to using amino acids and peptides in the urinary tract. These conclusions strengthen the notion that flexibility or modular components in central metabolism are important for UPEC replication and colonization in unique host niches.

## Materials and methods

### Bacteria and growth conditions

*E. coli* CFT073 was isolated from the blood and urine of a patient with acute pyelonephritis [53]. Bacteria was routinely cultured in lysogeny broth (LB) medium (per liter; 0.5 g NaCl, 10 g tryptone, 5 g yeast extract). For *in vitro* growth experiments, wild-type and mutants were cultured in MOPS defined medium [54] containing either 0.2% (w/v) glucose or 0.2% (w/v) glycerol as the sole carbon source. Wild-type CFT073 and mutants were examined for growth in pooled human urine collected from at least 4 healthy females and filter sterilized. LB medium and human urine cultures were inoculated 1:100 and defined medium cultures were inoculated 1:50 from overnight LB medium bacterial cultures and incubated with aeration at 37˚C. For *in vitro* iron-limitation experiments, bacteria grown in MOPS defined medium containing iron and 0.2% glucose were washed prior to being sub-cultured 1:50 into MOPS defined medium containing 0.2% glucose without iron. *In vitro* growth was examined in iron deplete and replete MOPS defined medium containing 36 μM $FeCl_3$. Growth curves were performed in triplicate and $OD_{600}$ was recorded every hour.

### Construction of mutants

*E. coli* CFT073 deletion mutants (Table 1) were constructed using the lambda red recombinase system [55]. Primers homologous to sequence within the 5' and 3' ends of the gene to be targeted were designed to replace the gene with a nonpolar kanamycin-or chloramphenicol-resistant cassette amplified from template plasmid pKD4 or pKD3, respectively [55]. *E. coli* CFT073 mutants were confirmed by PCR amplification using primers flanking the target gene sequence and comparing gene product size to wild-type CFT073 PCR product size. To decipher negligible product sizes PCR reactions were digested with a restriction enzyme (New England Biolabs). The antibiotic resistance cassette of the *fumC* mutant was removed with plasmid pCP20 to unmark the strain for deletion of additional genes; *fumA* and/or *fumB*. Oligonucleotides used to construct and confirm the *fumA*, *fumB*, *fumAB*, *fumAC*, *fumBC*, *fumABC*, and *sdhBfrdA* mutant strains are listed in Table S1.

### Complementation of mutants

To perform *in vitro* and *in vivo* complementation experiments, complementation plasmids were designed by PCR amplifying *frdABCD*, *sdhB*, and *fumC* from wild-type CFT073 genomic DNA using Easy-A high fidelity polymerase (Agilent), digesting with either HindIII and SacI or SphI and NotI restriction enzymes (New England Biolabs), and cloning into pGEN-MCS [56]. Oligonucleotides used for complementation construct design are listed in Table S1. The sequence of pGEN-*frdABCD*, pGEN-*sdhB*, and pGEN-*fumC* was confirmed by sequence analysis prior to transformation into the *sdhBfrdA* double mutant, *sdhB* single mutant, and *fumABC* triple mutant, respectively.

### Experimental UTI

The fitness contribution of the CFT073 oxidative and reductive TCA pathway mutant strains during co-challenge competition was assessed in the CBA mouse model of ascending UTI [57,

**Table 1. Bacterial strains used in this study.**

| Strain or Plasmid | Genotype or description[a] | Source or reference |
|---|---|---|
| *E.coli* strains | | |
| CFT073 | Wild-type pyelonephritis isolate (O6:K2:H1) | [53] |
| *fim* L-ON | CFT073 ΔIRL, *fim* invertible element locked on | [59] |
| *fim* L-OFF | CFT073 ΔIRL, *fim* invertible element locked off | [59] |
| | CFT073 Δ*sdhB* (Kan$^r$) | [1] |
| | CFT073 Δ*frdA* (Cam$^r$) | [5] |
| | CFT073 Δ*sdhBfrdA*(Kan$^r$Cam$^r$) | This study |
| | CFT073 Δ*fumA* (Kan$^r$) | This study |
| | CFT073 Δ*fumB* (Cam$^r$) | This study |
| | CFT073 Δ*fumC* (Kan$^r$) | [5] |
| | CFT073 Δ*fumAB* (Kan$^r$Cam$^r$) | This study |
| | CFT073 Δ*fumAC* (Kan$^r$) | This study |
| | CFT073 Δ*fumBC* (Cam$^r$) | This study |
| | CFT073 Δ*fumABC* (Kan$^r$Cam$^r$) | This study |
| Plasmids | | |
| pGEN-MCS | Low copy number plasmid (Amp$^r$) | [56] |
| pGEN-*frdABCD* | pGEN-MCS digested with SphI and NotI | This study |
| | replacing MCS site with *frdABCD* (Amp$^r$) | |
| pGEN-*sdhB* | pGEN-MCS digested with SphI and NotI | This study |
| | replacing MCS site with *sdhB* (Amp$^r$) | |
| pGEN-*fumC* | pGEN-MCS digested with SphI and NotI | This study |
| | replacing MCS site with fumC (Amp$^r$) | |

[a]Kan$^r$, kanamycin resistance; Cam$^r$, chloramphenicol resistance; Amp$^r$, ampicillin resistance.

58]. Female CBA/J mice (6–8 week old; 20 to 22 g; Jackson Laboratories) were anesthetized with ketamine/xylazine and transurethrally inoculated with a 50 μl bacterial suspension of 2 x $10^8$ CFU per mouse using a sterile polyethylene catheter (I.D. 0.28 mm x O.D. 0.61 mm) connected to an infusion pump (Harvard Apparatus). *In vivo* co-challenges were performed with a bacterial suspension containing a 1:1 ratio of wild-type *E. coli* CFT073 and CFT073 antibiotic-resistant mutant in PBS or a 1:1 ratio of two CFT073 antibiotic-resistant mutants in PBS or a 1:1:1 ratio for three CFT073 antibiotic-resistant mutants in PBS. For neutrophil depletion experiments, mice were injected via the intraperitoneal route with 100 μl of sterile PBS alone or sterile PBS containing 150 mg/kg of cyclophosphamide at 96 h and 100 mg/kg at 24 h prior to infection. Blood samples were analyzed to confirm absence of neutrophils in cyclophosphamide treated mice. Input CFU/ml was determined by plating serial dilutions (Spiral Biotech) of the inoculum onto LB agar with and without antibiotic. Urine of infected mice was collected 48 h post infection by abdominal massage and plated for CFU/ml. Following subsequent euthanization, bladder and kidneys were aseptically removed, weighed, and homogenized (OMNI International) in 3 ml PBS, and appropriate dilutions were spiral plated on LB agar with and without antibiotic to determine the output CFU/g of tissue. Viable counts were enumerated using QCount software (Spiral Biotech) and CFU from antibiotic-containing medium (mutant CFU) were subtracted from the total CFU from plates lacking antibiotic to determine the number of wild-type bacteria. For co-challenge experiments, competitive indices (CI) were calculated by dividing the ratio of the CFU of mutant to wild-type recovered from each mouse following infection by the ratio of the CFU of mutant to the CFU of wild-type present in the input. CI data were log-transformed and analyzed by the Wilcoxon signed-rank test to

determine statistically significant differences in colonization (*P*-value <0.05). A series of Mann Whitney tests were used to analyze CI data of PBS control vs cyclophosphamide mice in each respective organ site. A CI>1 indicates that the mutant out-competes the wild-type strain and a CI<1 indicates that the mutant is out-competed by the wild-type strain.

### Anaerobic testing on phenol red

Overnight LB medium cultures of wild-type CFT073 and mutants were spotted (5 μl) onto LB agar containing 0.04 g/liter phenol red alone, phenol red and 0.2% (wt/vol) glycerol, phenol red and 0.2% (wt/vol) fumarate-0.2% (wt/vol) glycerol, phenol red and 0.2% (wt/vol) glucose, and phenol red or 0.2% (wt/vol) fumarate-0.2% (wt/vol) glucose. Plates were incubated under anaerobic (BD GasPak EZ Anaerobe) conditions at 37°C for 24 h and subsequently imaged. These plates were transitioned to aerobic conditions at 25°C and imaged at 6 h and 24 h. Strains were also spotted on phenol red agar plates for examination under aerobic conditions at 37°C and imaged at 6 h and 24 h. These plates were then transitioned to 25°C and imaged at 48 h post-inoculation.

### Hydrogen peroxide and acid resistance assays

For hydrogen peroxide assays, approximately $10^8$ CFU/ml of wild-type CFT073 or mutant was inoculated into LB medium containing either 0% $H_2O_2$, 0.03% $H_2O_2$, 0.3% $H_2O_2$, or 3% $H_2O_2$. For acid resistance assays, overnight cultures were washed in phosphate buffered saline and $10^7$ CFU was inoculated into 100 mM MES buffered LB (LB-pH 2.5, LB-pH 5.0, and LB-pH 7.0). Input and output samples were collected from each assay to perform 1-fold serial dilutions in a microwell plate and a total of 10 μl from each dilution was drip plated onto LB agar. All hydrogen peroxide assays were carried out under static conditions at room temperature and samples were collected at 10, 20, 30, 40, 50 and 60 min post-inoculation and acid resistance assays were carried out at 37°C and samples were collected at 1h post-inoculation. Following incubation of LB agar plates at 37°C for 18 h viable colonies were enumerated at countable dilutions. Assays were performed in triplicate and averaged. A 2-way ANOVA was run to determine statistical difference between CFT073 and mutant strains at each time point.

### Antibiotic susceptibility assay

Wild-type CFT073 and the oxidative and reductive TCA pathway mutant strains were examined in LB medium containing chloramphenicol 10 μg/ml, trimethoprim 5 μg/ml, tetracycline 10 μg/ml, streptomycin 5 μg/ml, ampicillin 5 μg/ml, ciprofloxacin 250 ng/ml. Samples were collected over the course of 180 minutes, diluted and plated on LB agar to determine CFU. Assays were performed in triplicate and averaged. For antibiotic susceptibility testing using antibiotic test strips, overnight LB medium cultures of wild-type CFT073 and all oxidative and reductive TCA pathway mutant strains were diluted 1:100 in Mueller-Hinton broth, except the *fumABC* triple mutant which was diluted 1:50, and grown shaking at 37°C for approximately 2–3 h to an $OD_{600}$ = 0.5. Cultures were adjusted to lowest OD with phosphate buffered saline and a total of 150 μl was spread plate onto Mueller-Hinton agar. Once dry, a Minimum Inhibitory Concentration (MIC) Test Strip (Liofilchem MTS) was placed face up in the center of the agar plate using sterile forceps. The following antibiotics were tested: streptomycin 0.064–24 μg/ml, ampicillin 0.016–256 μg/ml, and ciprofloxacin 0.002–32 μg/ml. Input CFU was quantitated by performing 1-fold serial dilutions in a microwell plate and a total of 10 μl from each dilution was drip plated onto LB agar. Plates were incubated at 37°C for 18 h aerobically or for 24h in an anaerobic chamber and the MIC on the strip where inhibited growth was observed was recorded. Assays were performed in duplicate and averaged.

## qPCR

For fimbrial and flagellar gene expression studies, overnight LB medium cultures of wild-type CFT073, oxidative and reductive TCA pathway mutant strains, type 1 fimbriae mutants *fim* L-ON and *fim* L-OFF [59] were back diluted (1:100) and grown shaking in LB medium at 37˚C for approximately 2–3 h to an $OD_{600}$ = 0.5–0.6. For RNA stabilization a 2:1 ratio (v:v) of RNA protect (Qiagen) was added to 250 µl of bacterial sample and spun down (Thermo Scientific, Legend Micro 21, RT, 5 min, 9,600 x *g*). For gene expression of fumarase genes during iron replete and deplete conditions, samples were collected from MOPS defined medium cultures with and without 36 µM $FeCl_3$ grown at 37˚C for approximately 4 h to an $OD_{600}$ = 0.5–0.6. For RNA stabilization a 2:1 ratio (v:v) of RNA protect (Qiagen) was added to 100 µl of bacterial sample and spun down (Thermo Scientific, Legend Micro 21, RT, 5 min, 9,600 x *g*). Subsequent RNA isolation was performed with the RNeasy Mini kit (Qiagen) followed by DNAse-treatment of RNA with the TURBO DNA-free kit (Invitrogen). PCR with primers to amplify housekeeping gene, glyceraldehyde-3-phosphate dehydrogenase (*gapA*), was performed to check for contaminating DNA. cDNA synthesis was performed on DNA-free RNA samples with the iScript kit (Bio-Rad). 5 ng of total cDNA was added to each reaction, qPCR was performed with SyberGreen PowerUp reagent (Invitrogen) with primers to amplify the following genes, *fliA*; RNA polymerase sigma factor for flagellar operon, *fliC*; flagellin assembly, *fimA*; type 1 fimbrial protein, *papA* and *papA2*; major pillin subunit A, and *fumA*, *fumB*, *fumC*; fumarase. *gapA* expression was used as the internal control to calculate relative expression values of experimental genes. The comparative threshold cycle ($C_T$) method was used to determine the relative fold-change compared to wild-type CFT073. Oligonucleotides used in the qPCR experiments are listed in Table S1.

## Motility assays

The swimming motility of wild-type *E. coli* CFT073 and mutants was observed on soft agar plates (10 g/L tryptone, 5 g/L NaCl, and 2.5 g/L Bacto agar). Overnight bacterial cultures grown shaking at 37˚C were normalized to an $OD_{600}$ = 1.0 in tryptone broth (10 g/L) and using a sterile inoculating needle stabbed into the center of the motility agar and incubated at 30˚C for 16 h. The diameter of swimming was measured for each strain. Experiments were performed in duplicate and repeated for a total of three biological replicates. Statistical significance was determined by one-way ANOVA with Dunnett's multiple comparisons test.

## Invertible element PCR assay

Pyelonephritis isolate CFT073, previously constructed type 1 fimbriae mutants with the promoter situated in the on orientation (*fim* L-ON) or situated in the off orientation (*fim* L-OFF) within the invertible element (IE) [59], and all oxidative and reductive TCA pathway mutant strains were incubated statically in LB medium at 37˚C for 18 h. Cultures were standardized to an $OD_{600}$ = 0.5. The IE PCR assay was conducted as described previously [60]. In brief, IE_ F (5'-AGTAATGCTGCTCGTTTTGC-3') and IE_R (5'-GACAGAGCCGACAGAACAAC-3') were used to amplify the 601-bp invertible element. PCR products were subsequently digested with *Sna*BI and electrophoresed on a 3% agarose-1× TAE (Tris-acetate-EDTA buffer) gel to determine the orientation of the *fim* promoter within the IE. Digestion of the IE PCR product with the *fim* promoter positioned in the on orientation results in a 403 bp and a 198 bp band while digestion of the IE PCR product with the *fim* promoter positioned in the off orientation results in a 440 bp and a 161 bp band. To determine the proportion of IE DNA with the *fim* promoter situated in the on and off orientation pixel intensities of the bands on the gel image calculated by Image Lab 6.0 Software (Bio-Rad) were examined. In brief, pixel intensities of both PCR product bands of the *fim* promoter positioned in the on or off orientation were

added together to find the total bacterial population with the *fim* promoter positioned in the on or off orientation, respectively. Each total bacterial population value (total *fim* on or total *fim* off) was then divided by the total sum of pixel intensity (total *fim* on and total *fim* off) to calculate the percent bacterial population with the IE containing the *fim* promoter positioned in the on or off orientation.

## Ethics statement

All animal experiments were performed in accordance to the protocol (08999–3) approved by the University Committee on Use and Care of Animals at the University of Michigan. This protocol is in complete compliance with the guidelines for humane use and care of laboratory animals mandated by the National Institutes of Health.

## Supporting information

**S1 Fig. Complementation with *fumC* restores the *in vitro* growth defect of *fumABC*.** The growth defect of triple mutant strain *fumABC* was complemented with pGEN-*fumC* following growth in (A) LB medium, (B) defined medium containing 0.2% glycerol, and (C) defined medium containing 0.2% glucose. $OD_{600}$ values were recorded each hour and the mean of three independent trails is plotted.
(PDF)

**S2 Fig. The *in vitro* growth defect of SDH is restored when complemented with *sdhB*.** The growth defect of single mutant strain *sdhB* and double mutant strain *sdhBfrdA* was complemented with pGEN-*sdhB* following growth in (A) LB medium and (B) defined medium containing 0.2% glycerol. $OD_{600}$ values were recorded each hour and the mean of three independent trails is plotted.
(PDF)

**S3 Fig. *fumC* expression is induced under iron-limitation.** (A) Wild-type CFT073 and single fumarase mutant strains were grown shaking in defined medium containing 0.2% glucose with and without 36 μM $FeCl_3$ at 37°C until $OD_{600} = 0.5$ for RNA isolation. qPCR was performed to examine gene expression of fumarase genes, *fumA*, *fumB*, and *fumC*. The comparative threshold cycle (CT) method was used to determine the relative $\log_2$ fold-change of each strain in iron-replete medium compared to iron-limitation.
(PDF)

**S4 Fig. *E. coli* CFT073 sensitivity to hydrogen peroxide.** (A) Wild-type CFT073 was examined in LB medium containing either 0%, 0.03%, 0.3%, or 3% $H_2O_2$ to determine its susceptibility resulting in at least three logs of killing. Samples were collected every 10 minutes, diluted, and plated on LB agar to determine CFU. (B) Wild-type CFT073 and the oxidative and reductive TCA pathway mutant strains were examined in LB medium containing 0.0% $H_2O_2$. Samples were collected every 10 minutes, diluted, and plated on LB agar to determine CFU.
(PDF)

**S5 Fig. The oxidative and reductive TCA pathway mutant strains are not sensitive to pH during growth in mildly acidic conditions.** Wild-type CFT073 and the oxidative and reductive TCA pathway mutant strains were inoculated at $10^7$ CFU/ml and incubated in (A) 100 mM MES buffered LB, pH 7.0, and (B) 100 mM MES buffered LB, pH 5.0, for 60 minutes, diluted, and plated on LB agar to determine CFU. The dashed line represents the intended input CFU/ml at time zero. Bars display the mean of three independent trials, error bars show SEM values.
(PDF)

**S6 Fig. Pharmacological depletion of neutrophils does not rescue fitness defects for *sdhB* and *fumC* mutant strains.** Mice were treated with PBS control (open symbols) or cyclophosphamide (closed symbols) at 96 h and 24 h prior to infection. Blood samples were obtained from both groups for analysis. Wild-type CFT073 and either the (A, B) *fumC* mutant strain or (C, D) *sdhB* mutant strain were mixed in a 1:1 ratio and transurethrally infected into CBA/J mice. Urine, bladder, and kidneys were harvested 48 h post-infection and CFU/ml urine or g tissue was measured (A, C). $Log_{10}$ CI was calculated for each individual mouse organ (B, D). Bars display the mean. CI>1 indicates a fitness advantage and CI<1 indicates a fitness defect of the mutant. Significant differences in colonization ($^{*}P<0.05$) were determined using the Mann Whitney test.
(PDF)

**S7 Fig. Loss of SDH results in increased susceptibility to ciprofloxacin and streptomycin.** Wild-type CFT073 and the oxidative and reductive TCA pathway mutant strains were examined in LB medium containing (A) ciprofloxacin 250 ng/ml (B) ampicillin 5 μg/ml or (C) streptomycin 5 μg/ml. Samples were collected over the course of 180 minutes, diluted and plated on LB agar to determine CFU. Each symbol represents the average CFU of three independent trials with each indicated strain.
(PDF)

**S8 Fig. Loss of FumC decreases susceptibility to bactericidal antibiotics under anaerobic conditions.** The Minimum Bacterial Concentration (MBC) of (A) ciprofloxacin 0.002–32 μg/ml (B) ampicillin 0.016–256 μg/ml or (C) streptomycin 0.064–24 μg/ml observed for wild-type CFT073 and the oxidative and reductive TCA cycle mutant strains were recorded following overnight incubation on Mueller-Hinton agar with antibiotic test strips in anaerobic chambers. The values shown are averages from two independent trials.
(PDF)

**S9 Fig. Motility and orientation of the type 1 fimbriae promoter in the oxidative and reductive TCA pathway mutant strains are independent of one another.** (A) For the invertible element (IE) assay, strains were grown statically in LB at 37˚C for 18 h and standardized to $OD_{600}$ = 0.5 prior to PCR amplification, subsequent digestion with *Sna*BI and running on a 3% agarose-1× TAE (Tris-acetate-EDTA buffer) gel. The size of PCR products representing on or off orientation of type 1 fimbriae (*fim*) promoter are indicated. (B) The relationship between motility and *fim* phase position was determined by plotting the motility diameter against the *fim* L-ON percentage of the population for each fumarase mutant and wild-type CFT073. Linear regression determined $R^2$ = 0.87. (C) The relationship between motility and *fim* phase position was determined by plotting the motility diameter against the *fim* L-ON percentage of the population for the succinate dehydrogenase and fumarate reductase mutants with wild-type CFT073. Linear regression determined $R^2$ = 0.70.
(PDF)

**S10 Fig. Gene expression of flagellar and fimbrial genes in oxidative and reductive TCA pathway mutant strains suggests altered gene regulation is not responsible for motility phenotype.** (A-E) Strains were grown in LB shaking at 37˚C until $OD_{600}$ = 0.5 for RNA isolation. qPCR was performed to examine gene expression of flagellar genes (*fliA* and *fliC*) and fimbrial genes (*fimA*, *papA*, and *papA2*). The comparative threshold cycle (CT) method was used to determine the relative $log_2$ fold-change compared to wild-type CFT073. Bars denote the mean of three independent RNA isolation events, error bars are SEM.
(PDF)

## Acknowledgments

We thank Taylor Mitchell for her assistance during experiments and Sara Smith, veterinary technician, for conducting the cyclophosphamide animal experiments.

## Author Contributions

**Conceptualization:** Christopher J. Alteri.

**Data curation:** Stephanie D. Himpsl, Allyson E. Shea, Christopher J. Alteri.

**Formal analysis:** Stephanie D. Himpsl, Allyson E. Shea, Christopher J. Alteri.

**Funding acquisition:** Harry L. T. Mobley.

**Investigation:** Stephanie D. Himpsl, Allyson E. Shea, Jonathan Zora, Jolie A. Stocki, Dannielle Foreman, Christopher J. Alteri.

**Methodology:** Stephanie D. Himpsl, Allyson E. Shea, Christopher J. Alteri.

**Supervision:** Christopher J. Alteri.

**Writing – original draft:** Stephanie D. Himpsl, Allyson E. Shea, Christopher J. Alteri.

**Writing – review & editing:** Stephanie D. Himpsl, Allyson E. Shea, Christopher J. Alteri, Harry L. T. Mobley.

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
