## [Decision Letter · Decision Letter 0]

14 Nov 2019

Dear Dr. Alteri,

Thank you very much for submitting your manuscript "Flexible Energy Metabolism and Respiratory Pathways are Key Contributors to E. coli Fitness during UTI" (PPATHOGENS-D-19-01673) for review by PLOS Pathogens. Your manuscript was fully evaluated at the editorial level and by independent peer reviewers. The reviewers appreciated the attention to an important problem, but raised some substantial concerns about the manuscript as it currently stands. These issues must be addressed before we would be willing to consider a revised version of your study. We cannot, of course, promise publication at that time.

We therefore ask you to modify the manuscript according to the review recommendations before we can consider your manuscript for acceptance. Your revisions should address the specific points made by each reviewer.

(1) A letter containing a detailed list of your responses to the review comments and a description of the changes you have made in the manuscript. Please note while forming your response, if your article is accepted, you may have the opportunity to make the peer review history publicly available. The record will include editor decision letters (with reviews) and your responses to reviewer comments. If eligible, we will contact you to opt in or out.

(2) Two versions of the manuscript: one with either highlights or tracked changes denoting where the text has been changed; the other a clean version (uploaded as the manuscript file).

Additionally, to enhance the reproducibility of your results, PLOS recommends that you deposit your laboratory protocols in protocols.io, where a protocol can be assigned its own identifier (DOI) such that it can be cited independently in the future. For instructions see http://journals.plos.org/plospathogens/s/submission-guidelines#loc-materials-and-methods

We hope to receive your revised manuscript within 60 days. If you anticipate any delay in its return, we ask that you let us know the expected resubmission date by replying to this email. Revised manuscripts received beyond 60 days may require evaluation and peer review similar to that applied to newly submitted manuscripts.

[LINK]

Sincerely,

Andreas J Baumler

Associate Editor

PLOS Pathogens

Xavier Nassif

Section Editor

PLOS Pathogens

Kasturi Haldar

Editor-in-Chief

PLOS Pathogens

orcid.org/0000-0001-5065-158X

Grant McFadden

Editor-in-Chief

PLOS Pathogens

orcid.org/0000-0002-2556-3526

Reviewer's Responses to Questions

**Part I - Summary**

Reviewer #1: In this study, Himpsl, Shea, et al. investigate the energy and central carbon metabolism of UPEC in the context of urinary tract colonization. Previous work had suggested that mutants unable to perform an oxidative TCA cycle are less fit in bladder colonization model, while mutants unable to perform a branched TCA cycle did not. Here, the authors provide an exhaustive and in-depth characterization of mutants defective in the conversion of malate to fumarate as an extension of this work. In their mouse model or urinary tract infection, lack of FumC activity was associated with a decrease in bladder and kidney colonization. They also report that the fitness defect generated in the absence of an oxidative TCA cycle can be compensated by mutations in the fumarate reductase. In vitro, susceptibility of mutants to exogenous stress (acid stress, peroxide stress) and motility defects correlate with the observed defects in the mouse model. These data suggest that defects in the TCA cycle lead to increased susceptibility to ROS and other antimicrobials produced from alterations in the activity and composition of the respiratory chain.

While we have a good understanding of the E. coli metabolism of under standard laboratory conditions, the metabolism associated with host colonization by pathogenic E. coli is understudied. As such, the topic of this manuscript should be of great interest to the field. The careful and comprehensive analysis of mutant phenotypes in vitro and in vivo is commendable –certainly a lot of hard work. These findings should be helpful for the interpretation of previous findings on the function of the central carbon metabolism in E. coli and other Enterobacteriaceae. At this point, I my only concerns are regarding the writing and the novelty of the overall conclusions, as outlined below.

Reviewer #2: This manuscript by Himpsl et al builds on two previous publications by the Mobley group that implicated enzymes within the TCA cycle as determinants of UPEC fitness during UTI. Specifically, the previous studies showed that the fumarate dehydrogenase FumC and the succinate dehydrogenase SdhB are important for UPEC fitness within the urinary tract, where deletion of the fumarate reductase FrdA increased fitness. The current study digs deeper into this system, showing that deletion of frdA counters the fitness defect observed when sdhB is mutated. Results are also presented showing that only fumC, among the three fumarate dehydrogenases encoded by UPEC, is important for UPEC fitness during UTI. A series of in vitro experiments suggest that the main defects associated with deletion of sdhB and fumC are due to increased oxidative stress, and perhaps altered membrane integrity. The data are robust and, for the most part, presented in a way so that even non-experts in metabolism can appreciate the findings. The work increases our understanding of UPEC fitness determinants, though a lack of clear mechanisms for the many phenotypes described and some overlap with previous studies may be viewed as problematic.

**Part II – Major Issues: Key Experiments Required for Acceptance**

Reviewer #1: 1. The writing is fairly dense to a point that it is hard to read and understand what the authors are trying to say. For example, in the abstract, the authors use gene symbols, enzyme names, and abbreviations. This is all technically correct and I understand what the authors want to convey, yet I fear that my trainees and readers outside the field, this is not easily accessible. The findings are really interesting, but this might be lost on most readers.

2. While the study provides a good framework for the interpretation of the in vivo phenotypes, it is difficult to understand what the authors have really found – what is the conceptual advance? It comes as no surprise that manipulation of the central metabolism will result in pleiotropic effects and that the field should be carful in designing experiments.

3. The in vitro experiments suggest a few testable hypotheses as to why certain mutants are attenuated in vivo. It would be important to take this correlation between in vitro phenotyping and in vivo disease modeling a step further and experimentally test these ideas. For example, the authors surmise that neutrophil-derived products might be a stressor of UPEC. One would predict that neutrophil depletion as well as pharmacological or genetic manipulation would rescue the observed E. coli mutant phenotypes.

Reviewer #2: 1. The title refers ‘flexible metabolism and respiratory pathways’ as key contributors to E. coli fitness during UTI. In the end, the two gene products that are the focus of much of the study, FumC and SdhB, are associated with oxidative stress or changes in membrane integrity. More direct evidence to back up these possibilities (i.e. using genetic reporters or chemical sensors) would greatly strengthen the paper and distinguish it from the authors previous work implicating the TCA cycle as a fitness determinant during UTI.

2. The authors suggest that the altered sensitivities of some of the mutant strains to antibiotics is associated with changes in respiration or altered electrochemical gradients across the bacterial membrane (Fig 10 and S6). Do the mutants display differential sensitivity to antibiotics under anaerobic conditions?

3. The Fim switch is nearly entirely OFF in the sdhR and fumC mutants (Fig 11C), yet fimA transcription is for the most part unaffected in these mutants (Fig S8E). The reason for this dis-connect is not clear. Please clarify.

**Part III – Minor Issues: Editorial and Data Presentation Modifications**

Reviewer #1: 4. The use of the term “fumarate dehydratase” is a bit unusual (there is no water removed from the fumarate molecule). This enzyme is typically called fumarase or fumarate hydratase as it (inter)converts fumarate plus water to malate.

5. Do lystates of the fumABC mutant lack fumarase activity? It is conceivable that other, uncharacterized genes might encode an alternative fumarase or that other hydratases exhibit moonlighting activity.

6. Fig. 1E and other figures: It is hard to see which symbols are open or closed.

Reviewer #2: 1. The paper could probably be shortened substantially to improve readability and clarity. As is, it is difficult to work through the mountain of data and text. Discussion is too much of a recap of the Results section and could be shortened.

2. Many of the reported phenotypes are modest (small effect sizes), yet the authors use words like “crippled” when referring to slightly attenuated mutant strains. This should be adjusted.

3. The fumC mutant lags behind the wild-type strain, but reaches the same OD as the wild-type strain (Fig 6) regardless of the addition of iron. On page 14, the text suggests that this is only seen when iron is added (lines 284-286). This is not the case, based on the data in Fig 6.

4. Some of the symbols/lines in some of the graphs are difficult to discern (e.g. Fig. 10B). These could be better distinguished to help the reader.

5. Why is 25C used for the phenol red agar plate assays?

6. This study and the previous two studies by the Mobely group on TCA-associated enzymes use a single UPEC isolate, CFT073. The impact and generality of this study would be elevated if some of the key findings were corroborated in one or more additional strains, such as an ST131 isolate.

PLOS authors have the option to publish the peer review history of their article (what does this mean?). If published, this will include your full peer review and any attached files.

Reviewer #1: No

Reviewer #2: No

---

## [Editor Report · Decision Letter 1]

5 Feb 2020

Dear Dr. Alteri,

We are pleased to inform you that your manuscript 'The Oxidative Fumarase FumC is a Key Contributor for E. coli Fitness under Iron-limitation and during UTI' has been provisionally accepted for publication in PLOS Pathogens.

Before your manuscript can be formally accepted you will need to complete some formatting changes, which you will receive in a follow up email. A member of our team will be in touch within two working days with a set of requests.

Best regards,

Andreas J Baumler

Associate Editor

PLOS Pathogens

Xavier Nassif

Section Editor

PLOS Pathogens

Kasturi Haldar

Editor-in-Chief

PLOS Pathogens

orcid.org/0000-0001-5065-158X

Michael Malim

Editor-in-Chief

PLOS Pathogens

orcid.org/0000-0002-7699-2064
---

## [Editor Report · Acceptance letter]

21 Feb 2020

Dear Dr. Alteri,

We are delighted to inform you that your manuscript, "The Oxidative Fumarase FumC is a Key Contributor for E. coli Fitness under Iron-limitation and during UTI," has been formally accepted for publication in PLOS Pathogens.

Best regards,

Kasturi Haldar

Editor-in-Chief

PLOS Pathogens

orcid.org/0000-0001-5065-158X

Michael Malim

Editor-in-Chief

PLOS Pathogens

orcid.org/0000-0002-7699-2064